# Spatially resolved photocatalytic active sites and quantum efficiency in a 2D semiconductor

Olivier Henrotte [1,2] ✉, Seryio Saris[1], Franz Gröbmeyer[1], Christoph G. Gruber [1], Ismail Bilgin[3], Alexander Högele[3,4], Naomi J. Halas [5,6,7,8], Peter Nordlander [6,7,8], Emiliano Cortés [1] ✉ & Alberto Naldoni [9] ✉

Identifying reactive sites and measuring their activities is crucial for enhancing the efficiency of every catalyst. Reactivity maps can guide the development of next-generation photocatalysts like 2D transition metal dichalcogenides, which suffer from low conversion rates. While their electrocatalytic sites are well-studied, their photocatalytic sites remain poorly understood. Using scanning photoelectrochemical microscopy, we spatially resolve the photoreactivity of $MoS_2$ monolayers, a prototypical 2D transition metal dichalcogenide, for redox reactions, including $H_2$ production from water. Aligned-unaligned excitation-detection measurements reveal that photogenerated holes and electrons exhibit distinct behaviors. Oxidation products localize at the excitation spot, indicating stationary holes, while photoreduction occurs up to at least 80 microns away, showing exceptional electron mobility. We also elucidate the photochemical reactivity according to the nature of the electronic excitation, showing that the internal quantum efficiency of strongly-bound A-excitons outperforms weakly-bound (free-carrier like) C-excitons across the flake. These findings offer novel guidance to rationally design 2D photocatalysts via engineering their optical and charge extraction abilities for efficient solar energy conversion.

Two-dimensional transition metal dichalcogenides (2D TMDs) are promising material platforms for optoelectronics, sensing, and quantum information technologies, showing significant potential for groundbreaking applications[1,2]. They also hold immense potential for photocatalysis, due to their earth-abundance, strong light-matter interaction, and high surface-to-volume ratio, that can host many active sites for chemical reactions[3–5]. 2D TMDs offer multiple approaches for modifying their optical, electronic and surface properties through structural and strain control[6], phase engineering[7], and heterostructure formation with other low-dimensional semiconductors and photonic antennas for complex functional architectures[8]. This extensive tunability provides a broad range of options for optimizing photocatalytic performance, which requires a delicate balance between their optical (e.g., photon absorption and

[1]Nanoinstitute Munich, Fakultät für Physik, Ludwig-Maximilians-Universität München, München, Germany. [2]Regional Centre of Advanced Technologies and Materials, Czech Advanced Technology and Research Institute, Palacký University Olomouc, Olomouc, Czech Republic. [3]Fakultät für Physik, Munich Quantum Center, and Center for NanoScience (CeNS), Ludwig-Maximilians-Universität München, Nanoinstitut München, München, Germany. [4]Munich Center for Quantum Science and Technology (MCQST), München, Germany. [5]Department of Chemistry, Rice University, Houston, TX, USA. [6]Department of Electrical and Computer Engineering, Rice University, Houston, TX, USA. [7]Department of Physics and Astronomy, Rice University, Houston, TX, USA. [8]Technical University of Munich (TUM) Institute for Advanced Study (IAS), Garching, Germany. [9]Department of Chemistry and NIS Centre, University of Turin, Turin, Italy. ✉e-mail: o.henrotte@lmu.de; Emiliano.cortes@lmu.de; alberto.naldoni@unito.it

charge carrier dynamics) and catalytic properties (e.g., active sites and reactivity)[9,10].

Despite their potential in photocatalysis, most recent studies on 2D TMDs have focused on multicomponent designs, as co-catalysts to improve charge separation[11,12], and to enhance chemical reactions[13,14]. However, progress has been limited due to an incomplete understanding of their photocatalytic reactive sites and their activities. Instead, material design has relied on insights from the electrocatalytic properties of 2D TMDs (e.g., hydrogen evolution reaction)[15,16], where detailed global and local investigations emphasize the significance of edge sites[4,17,18], sulfur vacancies[6,19], and semiconductor-to-metal phase transitions as reactive sites[7,20,21]. However, electrocatalysis does not capture photoinduced processes such as exciton generation, charge separation and transport, or photocarrier-induced strain fields, all factors that can significantly influence photocatalytic reactivity. Therefore, it is essential to identify active sites and monitor their reactivity under appropriate operando conditions, using advanced approaches that can probe these aspects microscopically. A recent study highlighted the power of such operando approaches by revealing the local MoS2 photoreactivity according to the layer number[22]. These insights are critical for providing a deeper understanding that will lead to novel photocatalyst design, fully leveraging the tunability of 2D TMDs to develop next-generation materials and set new benchmarks in efficiency and sustainability[8].

To accomplish this goal, we employ scanning photoelectrochemical microscopy (SPECM) and identify the spatial distribution of reactive sites for oxidation and reduction on a model 2D TMD, a MoS2 monolayer (ML), using light as the sole external driving force. We observe and discuss the effects of the material's excitonic nature, electron/hole transport, and lateral confinement of photocarriers on chemical reactivity. Recent imaging techniques, such as surface photovoltage and fluorescence microscopy, have provided plenty of valuable insights into the microscopic excited-state dynamics and photochemical reactivity of a variety of catalysts[23–30]. However, these methods offer extreme spatial resolution but lack chemical specificity. In contrast, SPECM enables direct quantum efficiency mapping with high spatial resolution ( ~200 nm) and allows local quantification of redox reactions (e.g., $H_2$ evolution) under light excitation, providing a more direct assessment of catalytic performance.

## Results

### Physical characterization of monolayer MoS2

The ML-MoS2 flakes in the semiconducting phase were grown by a chemical vapor deposition (CVD) system previously reported (Methods) on a 285 nm $SiO_2$/Si substrate[31]. The photoluminescence (PL), Raman and optical images display the triangularly shaped CVD-grown MoS2 (Fig. 1a–c).

Characteristic properties of monolayers were verified by PL (Fig. 1d) and Raman (Fig. 1e) measurements[32]. The PL emission ( ~680 nm) of ML-MoS2 shifts and decreases in the presence of defect sites (e.g., corner and edge) as evidenced by the PL map contrast between the basal plane and the edges (Fig. 1a). A similar effect was observed in the presence of a multilayer, which in some cases can be observed in the middle of the MoS2 flake (Fig. S1)[33,34].

Raman spectroscopy was employed for thickness determination of the MoS2 flakes. The $E_{2g}$ (in-plane) and $A_{1g}$ (out-of-plane) vibrational modes serve as reliable indicators of layer thickness, as their frequencies shift systematically with the number of layers due to changes in interlayer interactions and bonding[35–39]. A peak difference ($\Delta_{peak} = E_{2g} - A_{1g}$) of ~19 cm$^{-1}$ was observed along the flake in our sample (Fig. 1e), also consistent with ML-MoS2[36,40]. The typical Raman peak wavelength shifts for multilayer MoS2, observed in the center of some flakes, corresponding to bilayer ( ~21.5 cm$^{-1}$) and multilayer ( ~24 cm$^{-1}$) MoS2 (Fig. S2). These shifts, together with the contrast observed in the optical images, enabled a fast identification of the presence of ML-

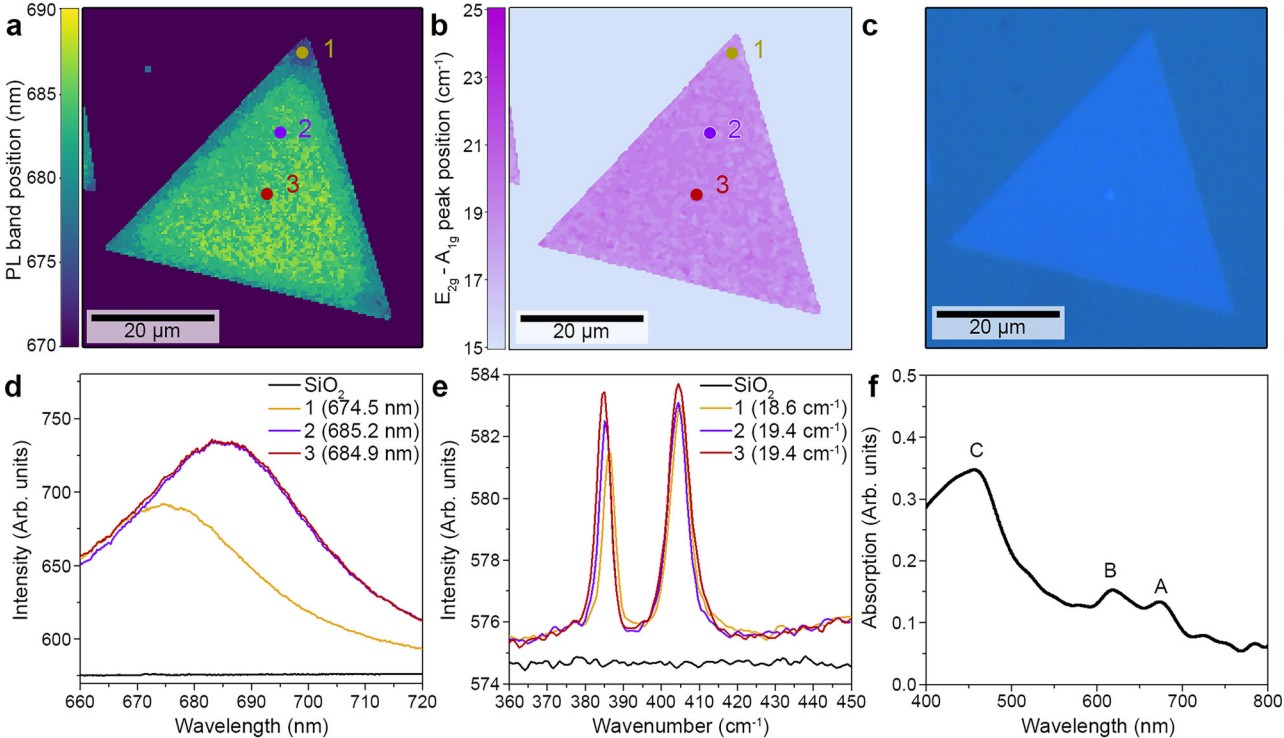

**Fig. 1 | Microscopic optical characterization of a monolayer MoS2 flake. a** PL band position map. **b** Raman peak difference ($\Delta_{peak} = E_{2g} - A_{1g}$) map. **c** Optical image of the flake shown in (**a**) and (**b**). **d** PL spectra for $SiO_2$ (black solid line) and position 1 to 3 (yellow, purple, and brown solid lines, respectively) indicated in (**a**) (circles of respective colors). **e** Raman spectra for $SiO_2$ (black solid line) and position 1 to 3 (yellow, purple and brown solid lines, respectively) indicated in (**b**) (circles of respective colors). **f** Absorption spectrum evidencing the A, B, and C peaks.

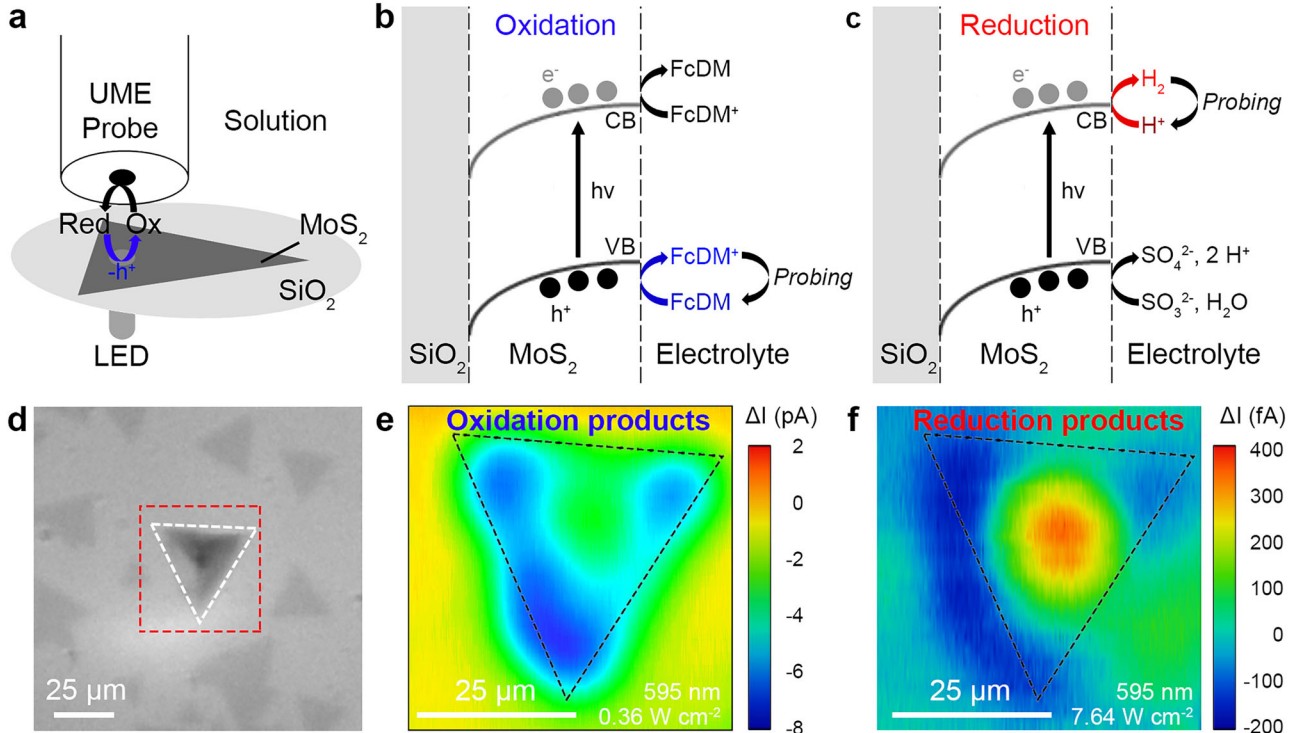

**Fig. 2 | Photocatalytic maps of oxidation and reduction products for monolayer MoS₂. a** Scheme depicting the aligned SPECM measurements performed on ML-MoS₂ for the photo-oxidation reaction. Illustrations describing the substrate generation – tip collection mode employed in this study for detecting the photocatalytic (**b**) oxidation and (**c**) reduction products. **d** Optical image of ML-MoS₂ deposited on SiO₂ with the dashed red square corresponding to the investigated area in (**e**) and (**f**). **e, f** Photoactivity maps of a ML-MoS₂ (black dashed triangle) corresponding to the highlighted flake in (**d**) for oxidation (**e**) and reduction (**f**) products. Conditions: $r_T = 5\,\mu m$, $RG = 50$, $v_{scan} = 5\,\mu m\,s^{-1}$, $r_{light} = 5\,\mu m$, with $E_T = -0.1\,V$ vs Ag/AgCl and $Z = 20\,\mu m$ for (**e**), and $E_T = 0.1\,V$ vs Ag/AgCl and $Z = 5\,\mu m$ for (**f**). The experiments were performed in aqueous solutions containing 1 mM FcDM and 0.1 M KCl (**e**) and 0.1 M Na₂SO₄ and 10 mM Na₂SO₃ (**f**).

MoS₂, indicating only a small multilayer region to be present at the center of the flake (Fig. 1c).

The local absorption of the as-grown ML-MoS₂ was obtained using scanning spectrophotometer microscopy[41]. The typical MoS₂ optical transitions were identified (Fig. 1f) as the A-transition (~670 nm; ~1.85 eV), B-transition (~620 nm; ~2.00 eV) and C-transition (~455 nm; ~2.72 eV). According to their optical transitions, the nature of the photogenerated charge carriers in ML-MoS₂ are known to be excitonic (~1.8 and 2 eV for the A and B excitons) or nearly-degenerate exciton states (~2.8 eV, namely, the C transition)[42,43].

**Spatially resolved photocatalytic reactive sites in monolayer MoS₂**

After identifying the physical properties of the ML-MoS₂, we proceeded to measure the spatial variation of its photocatalytic activity and determine the most reactive sites. To identify the photocatalytic reactive sites, we performed spatially-resolved aligned (excitation and probing at the same spot) SPECM measurements to detect the products generated at the MoS₂-liquid interface following oxidative and reductive processes. This method adopts an electrochemical probe (e.g., ultramicroelectrode, UME) that detects the concentration evolution of molecules near the MoS₂ surface[20,44,45].

The effect of pure photocatalytic events occurring at the solid-liquid interface on the species in solution is connected to the differential current measured at the UME under light illumination and dark conditions[41]. This photoactivity ($\Delta I = I_{T,\,Light} - I_{T,\,Dark}$) provides quantitative information on the local photoinduced redox reactions at specific excitation wavelengths (e.g., by exciting A, B, or C transitions) and for rather low power densities (<1 W cm⁻²).

We employed the substrate generation – tip collection mode (SG-TC) (Fig. 2a, details in Methods). In this modality, the probe, biased at a potential to selectively collect the chemical of interest[46], electrochemically detects the oxidized (negative $\Delta I$) or reduced (positive $\Delta I$) products generated from the photocatalytic reactions occurring at the ML-MoS₂-liquid interface (Fig. 2b, c). The area of interest, i.e., the ML-MoS₂ flake, was determined by optical microscopy (Fig. 2d).

For mapping the photo-oxidation activity, a redox mediator (i.e., ferrocene dimethanol, FcDM, Fe(C₅H₄CH₂OH)₂) featuring a single electron outer-sphere mechanism was employed (Fig. 2b, Eq. S1-S3)[41]. Fig. 2e shows the photo-oxidation activity map highlighting the highest photoactivity (~-7 pA), which was measured at the corners of the ML-MoS₂ flake.

Next, the reduction efficiency of the photogenerated electrons was evaluated through H₂ evolution from water (Fig. 2c, Eq. S4-S6). In this case, the opposite trend was observed from the photoreduction map, e.g., the highest photoactivity (~0.5 pA) was measured above the basal plane of ML-MoS₂ (Fig. 2f). It appears that the photo-oxidation and photoreduction processes take place in different areas of the MoS₂ flake.

The most abundant defects in MoS₂, sulfur vacancies and edge terminations, are well-known active centers in electrocatalytic processes such as the hydrogen evolution reaction (HER)[15,17,18,47–51]. Furthermore, extensive theoretical calculations have demonstrated that the electronic structure of MoS₂ for (photo)-electrocatalytic HER is highly dependent on its phase and atomic configuration[52–55]. However, little is known about the role of these defects in photocatalysis under excited-state conditions, where additional factors beyond static defect chemistry come into play. The obtained results deviated from the established understanding that MoS₂ exhibits catalytic activity at defect sites, specifically at the edges, for HER. Also, the measurement method itself may influence the observed activity as the proximity of the probe to the surface could introduce

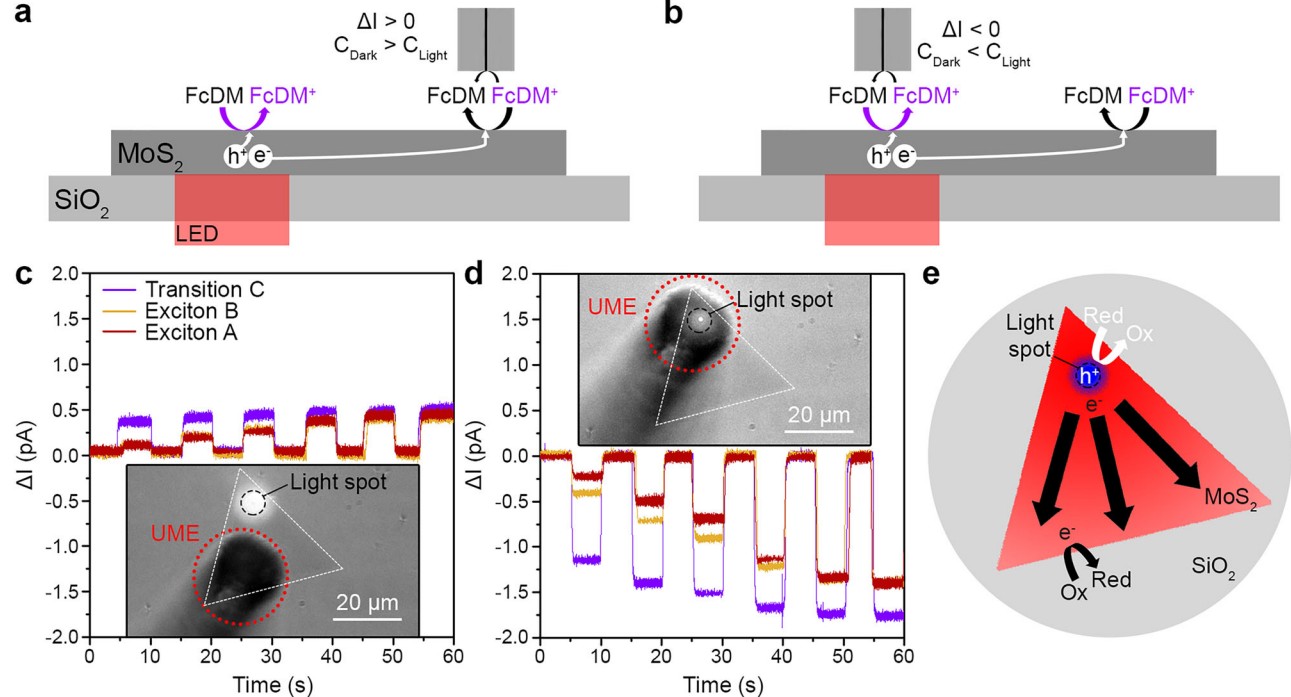

**Fig. 3 | Aligned-unaligned excitation-detection experiments in monolayer MoS₂.** Illustrations depicting the measurements performed with the nanoprobe when the excitation – detection was unaligned (**a**), or aligned (**b**). Photoactivity ($\Delta I$) measured far from the light excitation spot (**c**), and aligned with the light excitation spot (**d**), while the ML-MoS₂ was excited under chopped light (5 s; increased light power at every pulse; transition C, 455 nm, purple; exciton B, 595 nm, yellow; exciton A, 660 nm, dark red). The probe positions correspond to the respective inset image, showing the monolayer (ML-) MoS₂ flake (white dashed triangle), the position of the light excitation spot (black dashed circle), and the ultramicroelectrode (UME) position (red dashed circle). **e** Scheme representing the spatial distribution of the photo-redox processes detected at the UME in ML-MoS₂ when excited at a fixed position, with the corresponding photogenerated charge carriers (oxidation: holes, blue region; reduction: electrons, red region). Conditions: $r_T = 0.1\,\mu m$, $RG = 25$, $r_{light} = 5\,\mu m$ and $Z = 5\,\mu m$ for (**c**), (**d**). The experiments were performed in aqueous solutions containing 1 mM FcDM and 0.1 M KCl. FcDM⁺ is present in low concentration due to the redox couple equilibrium in water (Fig. S7). UME: ultra-microelectrode.

a hindering effect on the diffusion of the species from the MoS₂ to the solution[56].

Taking into consideration the influence of the probe during our measurements (i.e., impeding the mass transport of the produced species)[56], we suggest that the probe itself hindered the diffusion of the generated products from the MoS₂ flake (see Supplementary Section 1). This hindering effect decreased the apparent photoactivity at the basal plane for the detection of oxidized products (Fig. 2e, Fig. S3), while it increased the concentration of reduced products during the photoreduction map at the same location (Fig. 2f, Fig. S4).

To verify the distribution of the reactive sites along the MoS₂ flake, we performed spatially-resolved investigations with aligned and unaligned excitation-detection positions to determine the underlying mechanism behind our observations (Fig. 2e, f). The measurements were performed with a nanoprobe at a distance ($Z = 5\,\mu m$) larger than two times the radius of the probe insulating part ($r_{ins}$) avoiding the hindering effect of the probe on the diffusion of species. The FcDM/FcDM⁺ redox couple was used to decipher the redox reaction occurring at the probe position. Thus, the probe was positioned away from the excitation spot (Fig. 3a) or, alternatively, aligned with it (Fig. 3b). The oxidation (reduction) process occurs when the photogenerated hole (electron) is extracted from the MoS₂ by the surrounding molecules at the MoS₂-liquid interface. As such, a negative (positive) photoactivity should be observed when the initial concentration ($C_{Dark}$) of redox mediator (FcDM⁺) increases (decreases), accordingly to the reaction occurring at the MoS₂-liquid interface under illumination ($C_{Light}$).

Interestingly, the measured photoactivity was positive (reduction) when the probe was away from the excitation spot (Fig. 3c), but always negative (oxidation) when the probe was aligned with the light spot (Fig. 3d). In both cases, a rapid steady-state was observed revealing no influence from diffusion of the species[57]. This shows that the photogenerated charge carriers were extracted at the reactive sites in accordance with the observed redox process: holes at the light excitation spot, and electrons away from the light excitation spot (Fig. 3e). Since the sign of $\Delta I$ remained consistent across all measurements at a single probe position, this suggests that, regardless of the excited optical transitions, the holes are relatively static, while the electrons exhibit significantly higher mobility, traveling through the ML-MoS₂. These results provide insight into our previous observations for the aligned measurements, which only detected the products from photo-oxidation (Fig. 2e) or photoreduction (Fig. 2f) diffusing to the probe. This confirmed a significant charge carrier separation, where hole extraction is relatively localized at the excitation area, and electrons exhibit high mobility throughout the ML-MoS₂ (Fig. S4b). Furthermore, this corroborates our discussions on the hindrance of the probe on the diffusion of photoproducts from the MoS₂ to the probe (Supplementary Section 1, Fig. S3-S6), reinforcing the spatially selective nature of the observed photo-induced processes (Fig. 3e).

The observed photoactivity showed a dependence on the incident light intensity and a selectivity for optical transition: transition C > exciton B ≈ exciton A (Fig. 3c, d). The intense photoactivity observed at lower power density for the C-transition compared to B/A-excitons (at least two times higher) is due to its higher optical absorption (0.35 at 455 nm vs 0.13 at 595/660 nm) and to the nature of the generated charges: more free carriers are generated from C-transition excitation[58]. This finding shows that the nature of the generated charge carriers influences their mobility along the ML-MoS₂.

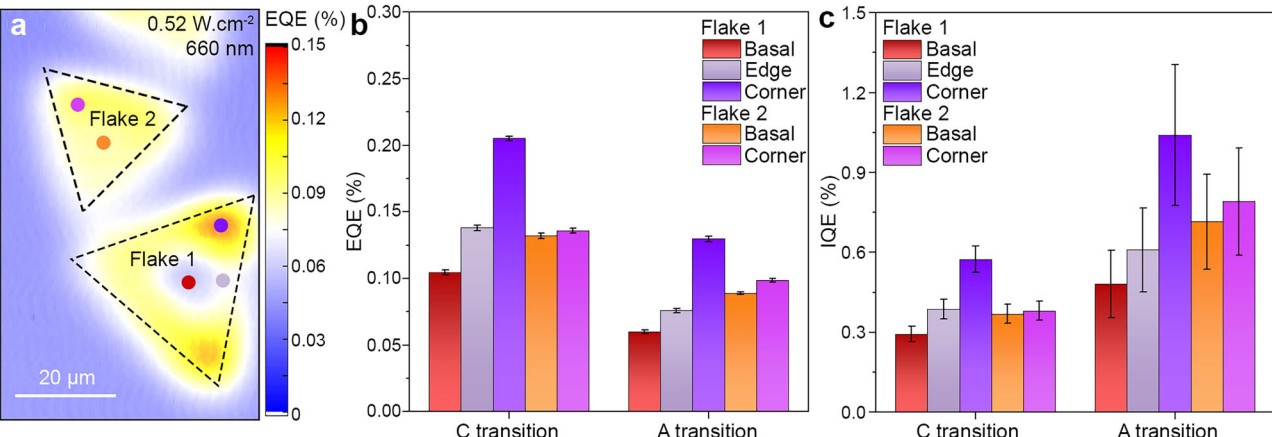

**Fig. 4 | Spatially-resolved quantum efficiency of monolayer MoS₂. a** External quantum efficiency (*EQE*) map of monolayer (ML-) MoS₂ flakes excited at $\lambda_{660nm} = 0.52\,W\,cm^{-2}$ with $r_{light} = 5\,\mu m$. The dashed triangles correspond to the highlighted flakes in Fig. S12. **b** External and (**c**) internal quantum efficiency at different positions (Flake 1 basal: dark red; edge: gray; corner: purple; Flake 2 basal: orange; corner: light purple) inside the ML-MoS₂ according to the excited transition at $-1.2 \times 10^{12}$ photons s⁻¹ (C transition at 455 nm; A transition at 660 nm). The error bars represent the standard deviation.

Based on the unaligned measurements, we conclude that the charge carriers are laterally confined inside the ML-MoS₂ because of (1) the surface of the MoS₂ flake (~728 μm²) and (2) the extraction pathways for the generated electrons. The local concentration of FcDM⁺ limits electron extraction (Fig. S7), which explains the $\Delta I_{Max}$ of 0.5 pA (Fig. 3c). Once this limitation is reached along the whole flake, the highest photoactivity is observed ($\Delta I_{Min}$ of −1.7 pA, Fig. 3d), which is related to the hole extraction. Additional information regarding the electrolyte (FcDM/FcDM⁺ concentration) influence on the observed photoactivity are available in Supplementary Section 2 (Figs. S8-S10).

We subsequently performed the same experiment on a larger MoS₂ flake (~3495 μm²) to independently confirm the observed carrier transport and extraction at the reactive sites (Fig. S11a). The surface of MoS₂ was increased by 4.7-fold in this larger sample, compared to the previous smaller flake. Here, one would anticipate a similar increase in the $\Delta I_{Min}$ for the aligned measurement, as the MoS₂-liquid interface determines the electron extraction. The obtained $\Delta I_{Min}$ for the larger flake (Fig. S11b) was ~4 times that of the smaller flake, confirming our initial interpretation. Moreover, the $\Delta I_{Max}$ values were still ~0.5 pA, consistent with the limited amount of FcDM⁺ available in solution.

Finally, the probe signal at ~20 μm (position 2 in Fig. S11) and ~80 μm (position 3 and 4 in Fig. S11) away from the light spot showed a dependence on photoactivity with distance from the excitation light spot and on its intensity. The photoactivity decreased with increasing distance from the excitation spot, and increased with light intensity until reaching a maximum, determined by the MoS₂-liquid interface enabling the free electrons to interact with FcDM⁺ (Fig. S11b). The photoactivity evolution, observed even 80 μm away from the excitation spot, suggests a remarkable mobility of electrons inside the ML-MoS₂ under continuous wave illumination, in light of the previously reported diffusion lengths of excitons under mild conditions in PL measurements (<2 μm)[58–61]. Notably, the size of the ML-MoS₂ flake was the key parameter limiting the observed diffusion length in our experiments. Our discovery highlights the significance of the photon flux on the diffusion length of the generated electrons. Thus, using a precise photon flux would allow to drive the reduction reaction at a specific distance from the excitation spot.

## Quantum efficiency of monolayer MoS₂ photoreactive sites

Following our understanding of the spatial distribution of the reactive sites within ML-MoS₂, we determined their efficiency for photocatalytic activity.

Since the photo-oxidation at the excitation location is directly linked to the ability of the electrons to interact with the solution, we conducted SPECM aligned experiments for the photo-oxidation reaction at different location of the ML-MoS₂ (Fig. 4a) to investigate its wavelength-dependent quantum efficiency (QE). This corresponds to the ability of the photocatalyst to produce chemicals according to the incident light exciting the catalyst, critical for correlating the photoactivity with the optical transitions.

The measured photoactivity normalized by the incident photon flux and energy of the ML-MoS₂ gives the local external quantum efficiency (EQE) (Fig. 4a, see Methods)[41]. Then, the internal quantum efficiency (IQE) was calculated by dividing the EQE by the ML-MoS₂ absorption (Fig. 1f). The former provides information about the catalytic efficiency (e.g., number of charge carriers/molecules produced per photon), while the latter reveals the material's ability to efficiently generate charge carriers, inducing a chemical reaction. In combination, EQE and IQE provide crucial metrics on the performance of the photoactive material.

The marked positions in Fig. 4a correspond to the positions where the measurements were performed. The EQE (Fig. 4b) and IQE (Fig. 4c) follow a similar trend for each investigated wavelength: $QE_{corner} > QE_{basal\,plane}$, in agreement with the defect sites showing superior reactivity (Fig. 1a). The increase in QE at the corners are ~1.7 times the QE at the basal plane for Flake 1, and almost no enhancement is observed for Flake 2, which can be due to the difference in superficies relative to the light excitation and the flakes. As a multilayer center was observed in Flake 1 (Fig. S12), we briefly discussed the influence of the flake morphology on the QEs in Supplementary Section 3 (Fig. S13-16). The EQE obtained for transition C (0.21%) is ~1.6 times larger than the EQE for exciton A (0.13%) at the corner of the flake. Similarly, the EQE at the basal plane is ~1.7 times greater for C compared to A. This is in agreement with a recent report by Austin et al., demonstrating the rapid extraction of free carriers from a photoelectrochemical cell for the excitation of the C transition[58].

Strikingly, the IQE observed here shows the opposite trend (1.04% for exciton A; 0.57% for transition C at the MoS₂ corner). These results highlight an essential discovery in our experiments, namely that bound excitons show the highest IQE observed in this system. A possible explanation is that the redox mediator allows for exciton charge transfer between MoS₂ and the solution, which dissociates the exciton and generates free electrons accumulating in the ML-MoS₂[62,63]. This finding suggests that optimizing the optical absorption for the

generation of bound excitons should significantly boost $MoS_2$ photoactivity.

To understand the effect of the lateral confinement shown in Fig. 3 on the QE, we evaluated the EQE and IQE at the basal plane and at the corner for low ($\sim 1.2 \times 10^{12}\,s^{-1}$) and high photon fluxes ($\sim 14 \times 10^{12}\,s^{-1}$) according to the excited transition (Fig. S14).

The previous trend is still observed (EQE: C-transition > A-exciton; IQE: A-exciton > C-transition), becoming more pronounced at high photon fluxes than at low photon fluxes. However, the EQE and IQE values for high photon fluxes decrease significantly, consistent with lateral confinement of the electrons inside the small $MoS_2$ flake.

Interestingly, the QE for the A-exciton shows the lowest decrease with increasing photon flux ($\sim 1.6$ to 2.8-fold), which indicates that lateral confinement is reached faster for the C-transition, suggesting a higher recombination rate of the free charge carriers. This can be due to the excitonic nature of the A-transition (bound excitons), with its higher binding energy compared to the nearly degenerate exciton produced at the C-transition, which requires little to no energy to dissociate (free carriers)[64,65].

## Discussion

In conclusion, we successfully mapped the photocatalytic activity for oxidation and reduction reactions occurring at the monolayer $MoS_2$-liquid interface, demonstrating the possibility of producing $H_2$ at the $MoS_2$ without an external bias or co-catalyst. A clear spatial separation of photocatalytic reactive sites was observed, and linked to the illumination position. This highlights the critical role of the location of light excitation in driving localized oxidation due to the static nature of photogenerated holes, and the importance of light intensity in determining the distance for reduction processes, due to the remarkable mobility of the electrons. Furthermore, the available surface directly impacted the charge extraction at the interface due to lateral confinement, and consequently also impacted the photocatalytic efficiency. This demonstrates the relevance of fabricating larger 2D materials with superior anisotropy for boosting their quantum efficiency.

These findings reinforce that, while defects contribute to reactivity, photocatalytic efficiency is governed not only by local defect chemistry but also by optically driven excited-state properties that extend beyond the well-studied behavior of $MoS_2$ under electrocatalytic conditions. Crucially, by employing spatially resolved SPECM measurements, we reveal that oxidation and reduction processes are not uniformly distributed across the flake but instead correlate with distinct transport behaviors of photogenerated carriers.

It is quite notable that the bound excitons in this system showed the highest internal quantum efficiency compared to free carriers. This can be explained by the higher-energy transition and the lower binding energy of carriers generated at the C-transition, leading to faster non-radiative recombination compared to the A-transition. Our results provide crucial insight for developing efficient photocatalysts based on monolayer $MoS_2$. This can be accomplished, for instance, by enhancing the absorption cross-sections using an underlying metasurface as a lens, and also by controlling reactive sites in 2D materials through defect engineering and the formation of van der Waals heterostructures. Moreover, our study also showcases the ability of SPECM in establishing microscopic structure-function relations, which can be directly relevant for deepening our understanding of these effects in a wide range of novel semiconductors, paving the way for innovative advances in photocatalysis.

## Methods

### Monolayer $MoS_2$ preparation

$MoS_2$ monolayers were synthesized using $MoO_2$ and S powders (Sigma Aldrich, 99% and 99.998% metals basis, respectively) as precursors through the vapor phase chalcogenization method in a three-zone furnace CVD system (Carbolite Gero) with a 1-inch quartz tube[31]. An aluminum boat containing $MoO_2$ powder was placed at the center of the first heating zone, with a 285 nm $SiO_2$/Si substrate suspended face-down directly above it. A separate crucible filled with S powder was positioned 20 cm upstream from the center. The temperature was gradually increased to a growth temperature of 800 °C at a rate of 5 °C min$^{-1}$ under a 200 sccm Argon flow, and maintained at that temperature for 15 min before cooling down to room temperature.

### Physical characterization of monolayer $MoS_2$

Photoluminescence and Raman measurements were performed with a WiTec optical microscope with a 50x objective (Zeiss EC Epiplan-Neofluar HD Dic 50x / 0.8). For spectral acquisition, the microscope was equipped with CW lasers (PL: 633 nm, 2 mW, 0.3 s integration; Raman: 532 nm, 10 mW, 0.36 s integration). The data were processed to reduce noise levels by averaging and removing artificial peaks for cosmic ray removal.

The optical properties were obtained using a spectrophotometer coupled with the positioning system of the SPECM (HEKA Gmbh, Lambrecht, Germany). Scanning spectrophotometer microscopy (SSM) consists of a white light source (Lambda DG 4 Xenon Arc bulb, Sutter instruments, USA) focused by an objective ($r_{beam} = 5\,\mu m$, LUCPLFLN40x, $NA = 0.6$, Olympus, Japan) illuminating the sample from below, aligned with the spectrophotometer via a glass capillary (100 μm radius) connected to an optical fiber (100 μm radius, $NA = 0.22$, Thorlabs, USA) above the sample platform to collect the transmitted light. The absorption values correspond to $-Log\,(T)$, where $T = I/I_O$, with $I$ and $I_O$ corresponding to the transmitted light through the measured $MoS_2$ sample and the bare substrate ($SiO_2$), respectively.

### Scanning photoelectrochemical microscopy measurements

SPECM experiments were performed with an ELP 3 SECM-FL (HEKA Elektronik GmbH, Lambrecht, Germany), a PG 618 USB bipotentiostat (HEKA Elektronik GmbH, Lambrecht, Germany) and fiber-coupled LED with different wavelengths: 455 ± 7, 595 ± 40, and 660 ± 9 nm (Thorlabs). Unless noted otherwise, the experiments were performed in KCl (0.1 M) and Fc(MeOH)$_2$ (1 mM). All chemicals were from Sigma-Aldrich, purchased at the highest available purity and used without further purification. The three-electrodes setup consisted of Pt microelectrodes (HEKA Elektronik GmbH, Lambrecht, Germany) employed as the working electrode (WE), a Leak-Free (LF-1-100, in KCl 3.4 M) electrode (Innovative Instruments, Inc., USA) used as the reference electrode (RE), and a Pt wire used as the counter electrode (CE). The sample under investigation was not connected at the bipotentiostat. For the measured maps and aligned measurements, the light beam ($r = 5\,\mu m$) and the microelectrode were aligned prior to measurements.

The ultramicroelectrode (UME) was biased to measure the current at the diffusion plateau ($E_T = -0.1$ V vs RE for every experiments except for the $H_2$ production, where $E_T = 0$ V vs RE), which corresponds to the diffusion limited current, and is given by the following equations in solution (i.e., $I_{T,\infty}$) and near the substrate (i.e., $I_{T,ins}$) due to the hindering effect on the diffusion of both the substrate and the probe (corresponding to negative feedback)[66]:

$$I_{T,\infty} = 4nFCDr_T \qquad (1)$$

$$I_{T,ins} = 4nFCDr_T \beta(RG)Ni_T(L, RG) \qquad (2)$$

Where $n$ is the number of electrons involved; $F$ is the Faraday constant (96485.33 s.A/mol); $C$ is the concentration in Red/Ox species; $D$ is the diffusion coefficient of the Red/Ox species ($7.8 \times 10^{-6}$ cm²/s)[67]; $\beta(RG)$ is the $RG$ impact on the diffusion to the probe active area and $Ni_T(L)$, the hindering effect of the near surface on the probe current with $L$ corresponding to the ratio between $d$ and $r_T$.

The chopped light measurements on ML-MoS$_2$ were performed at different wavelengths (455 nm, transition C; 595 nm, exciton B; 660 nm, exciton A) with increasing light intensity (Table S1) for each subsequent 5 s duration of illumination (Fig. 3b). The measurements were conducted in the presence of FcDM/FcDM$^+$, with FcDM$^+$ present in low concentrations due to the redox couple equilibrium in water (Fig. S3).

The ML-MoS$_2$ local QE was determined from the chopped light measurements using a previously introduced COMSOL diffusion model simulating the substrate $\Delta I$ based on the probe $\Delta I$[41]. The diffusion model was simulated using the "Transport of Diluted Species" physics in a 2D asymmetric mode considering the substrate as an electrochemical object corresponding to the light beam size. The diffusion model provides information such as the ratio between the probe current and the substrate current (called collection efficiency, CE), the probe geometry influence and the effect of the probe-substrate distance on the CE. The external and internal quantum efficiencies were calculated as follow:

$$EQE\ (\%) = \frac{\Delta I_S/e}{P/h\upsilon} \qquad (3)$$

with

$$CE = \frac{\Delta I_T}{\Delta I_S} \qquad (4)$$

$$IQE\ (\%) = \frac{EQE\ (\%)}{Absorption} \qquad (5)$$

Where $\Delta I_T$ is the measured probe current related to the photo-products, $\Delta I_S$ corresponds to the calculated photocatalytic events effectively occurring at the surface, e is the electron charge ($1.602176634 \times 10^{-19}$ C), P is the power density of the incident light, h is the Planck constant ($6.62607015 \times 10^{-34}$ m$^2$ kg s$^{-1}$), $\upsilon$ is the photon frequency, and CE is the collection efficiency between the electrochemical probe and the investigated surface.

## Simulation model

A two-dimensional diffusion model was implemented in COMSOL Multiphysics (v6.1) using the transport of diluted species physics interface to simulate the steady-state spatial distribution of molecular hydrogen in aqueous solution, generated heterogeneously from a reactive surface. The objective was to evaluate local concentration gradients and the impact of mass transport limitations in the presence of a non-reactive structure mimicking our electrochemical probe.

The simulation domain measured 60 μm in width and 20 μm in height. A 40 μm-wide reactive surface was positioned along the bottom edge of the domain, centered horizontally. To represent the physical presence of the probe, a 50 μm-wide passive object was placed 5 μm above the surface, aligned with the center of the illuminated region. Hydrogen generation was considered under two distinct configurations: (i) localized generation within a 10 μm-wide region centered beneath the probe, corresponding to direct photoexcitation at the light spot, and (ii) distributed generation over the remaining 30 μm of the reactive surface, simulating activity occurring away from the illuminated region.

In each case, an arbitrary total flux was imposed and normalized over the respective generation zone to maintain consistent overall H$_2$ production. The diffusion coefficient of hydrogen was set to $4.8 \times 10^{-9}$ m$^2 \cdot$s$^{-1}$, and the initial concentration of H$_2$ was assumed to be zero throughout the domain.

## Data availability

All data supporting the findings of this study are included in the Article, Supplementary Information and source data files, including PL, Raman, SPECM measurements, as well as COMSOL 6.1 (https://www.comsol.com) simulation results (available on the following public depository: https://doi.org/10.5281/zenodo.15806414). Source data are provided with this paper.

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

## Acknowledgements
The authors acknowledge funding and support from the Deutsche Forschungsgemeinschaft (DFG, German Research Foundation) under Germany´s Excellence Strategy – EXC 2089/1 – 390776260, the Bavarian program Solar Technologies Go Hybrid (SolTech), the Center for NanoScience (CeNS) and the European Commission through the ERC SURFLIGHT. I.B. acknowledges funding from the Alexander von Humboldt Foundation. A.H. acknowledges funding by the European Research Council (ERC) under the Grant Agreement No. 772195, the Deutsche Forschungsgemeinschaft (DFG, German Research Foundation) within Germany's Excellence Strategy under grant No. EXC-2111-390814868, and the Bavarian Hightech Agenda within the EQAP project. P.N. and N.J.H. acknowledge support from the Robert A. Welch Foundation under grants C-1222 and C-1220. N.J.H., P.N., O.H., and E.C. acknowledge the Institute for Advanced Study (IAS) from Technische Universität München (TUM) for financing the focus group on "Sustainable photocatalysis using plasmons and 2D materials (Sus-PhuP2M)" as part of the Hans Fisher Senior Fellowships program. S.S. acknowledges the funding support of the Swiss National Science Foundation (SNSF), under Grant Number P500PN_202653. A.N. acknowledges the project CH4.0 under the MUR program "Dipartimenti di Eccellenza 2023-2027" (CUP: D13C22003520001) and the project GreenSWaP (Green Solar-to-Propellant Water Propulsion) funded by the European Union under the European Innovation Council grant agreement N° 101161583. E.C. acknowledges the Federal Ministry of Research, Technology and Space (BMFTR) in Germany for financial support.

## Author contributions
O.H., E.C., and A.N. conceived and designed the project. I.B. fabricated the samples. F.G. and C.G.G. performed the PL and Raman measurements. O.H. designed and performed the SSM and SPECM experiments, and the COMSOL simulations. E.C. and A.N. supervised the project. E.C., A.N., N.J.H., P.N., and A.H. financed the project. O.H. and S.S. wrote the manuscript. All authors discussed the results and edited the manuscript.

## Funding

## Competing interests
The authors declare no competing interests.
