## [Peer Review File · Nature Communications]

Spatially Resolved Photocatalytic Active Sites and Quantum Efficiency in a 2D Semiconductor

Corresponding Author: Professor Alberto Naldoni

Version 0:

Reviewer comments:

Reviewer #1

(Remarks to the Author)

1. The number of test samples is too small, and the test results are not universal. It is suggested to increase the number of test samples and increase the supporting points of the conclusions of the paper.
2. Lack of theoretical calculation, and lack of theoretical calculation support for photocurrent.
3. From the perspective of innovation, although your research topic has some exploration value, unfortunately, we think that the innovation shown in the current manuscript is not outstanding enough.
4. For Fig. 4c and Fig. S6b, the reviewer requests that the authors examine the distribution of the data points to identify any potential outliers or biases that may be contributing to the larger error bars.
5. The coordinates of the image are too small, which is not conducive to readers' reading. We suggest that the coordinates of the image be enlarged.

Reviewer #2

(Remarks to the Author)

In this manuscript, Henrotte et al. used a SPECIM-based method to study the spatial distribution of photogenerated charge carriers and their reactivities on a prototypical 2D semiconductor monolayer (ML-MoS₂). Using an interesting aligned-unaligned excitation-detection methods, the authors were able to clearly show that photooxidation is localized at the excitation spot of ML-MoS₂, while photoreduction occurs up to 80 microns away due to the exceptional mobility of the photogenerated electrons in ML-MoS₂. Overall, this work provides a valuable tool to study the photochemical reactivity of 2D TMDs, which is not yet fully understood, but several concerns also need to be addressed before publication. Therefore, I recommend this work to be published after the major revisions listed below.

1. Using the SPECIM method, the authors showed that the reactivities of the photogenerated electrons were localized at the basal plane of ML-MoS₂ (Figure 2f). This is quite contradictory to previous views that the active sites of 2D TMDs are mainly corners and edges. In this sense, the authors failed to spatially resolve the photocatalytic reactive sites of ML-MoS₂, the main target of this study. The authors must make this point clear and explicit in the manuscript, otherwise the validity of the presented method would be compromised.
2. The authors stated that they directly measured the spatial reactivity of photogenerated electrons through electrochemically detecting the H₂ products. However, this photochemical study was only carried out without external electrochemical bias or Pt cocatalysts, which means that H₂ production might be relatively difficult. It appears that no solid evidence has been provided for the formation of H₂ on an ML-MoS₂ flask and they unambiguously measured the H₂ products rather than other reductive substances in this study. The authors might need to further clarify this point.
3. Regarding the SPECIM method, did the authors strictly remove dissolved oxygen? In Figure 2b, what is the reduction product driven by the photogenerated electrons? Whether the photochemical reduction product would affect the SPECIM detection of FcDM⁺? Another question concerns the use of sulfite (SO₃²⁻) in measuring the photoreduction activities of ML-MoS₂ (Figure 2c). In this case, the authors used SO₃²⁻ to remove the photogenerated holes. Since the by-products, sulfite radicals, were well-known strong oxidants, this might destroy the structure of ML-MoS₂ and also affect the detection of the photoreduction activities of ML-MoS₂ (e.g., the consumption of photogenerated electrons by this radical would result in a much lower production yield of H₂). The authors are recommended to use other sacrificial reagents instead of sulfite anions, or check the influences of different sacrificial reagents.
4. The aligned-unaligned excitation-detection method is an interesting experiment. My concern is why the authors only use

the FcDM-KCl system (for the determination of hole activity) to detect the reactivities. In Figure 3b, which substance is oxidized with respect to the measured positive current? Is it necessary to perform both FcDM-KCl and Na₂SO₃-Na₂SO₄ measurements for each aligned or unaligned experiment? This will help readers better understand these experiments. 5. The authors emphasized the numerous advantages of SPECM over other surface photovoltage and fluorescence microscopies (Line 73-77). It is recommended to cite several important papers to help readers better understand this work, for example, Nature, 2022, 610, 296–301; Nature, 2016, 530, 77–80; Sci. Adv. 2021, 7, eabj4452; J. Phys. Chem. Lett. 2023, 14, 5410–5425.

Reviewer #3

(Remarks to the Author)

This paper by Henrotte et al. titled “Spatially resolved photocatalytic reactive sites and quantum efficiency in a 2D semiconductor” details HER at basal vs corner sites and as a function of photoexcitation distance. This is very exciting work by the authors. I would recommend publication after minor revisions.

Abstract:

“Oxidation products localize at the excitation spot, indicating stationary holes, while photoreduction occurs up to 80 microns away, showing exceptional electron mobility”. This distance is limited by the size of the triangle shown in the SI. This should be more clearly stated that the distance is limited by the size of the triangle and that is the upper limit in these sets of experiments. Also can the authors suggest from their data how far they think the electron could travel if it the triangle was larger.

“We also elucidate the photochemical reactivity according to the nature of the electronic excitation, showing that strongly-bound A-excitons outperform weakly-bound (free-carrier like) C-excitons across the flake.” This is true for IQE and not EQE. This should be specifically said in the abstract to not mislead the reader.

Figure 2b what is the reduction reaction to balance the oxidation reaction?

Figure 2 e and f – why are the oxidation products (holes) more favorable on the edges? I would have thought that the edges contain more sulfur vacancies and would trap electrons? It would be good to understand why the holes are trapped at the edges/or alternatively why the electrons stay at the basal sites.

Line 162 – 167 “Taking into consideration the influence of the probe.... The probe itself hindered the diffusion of the generated products at the edge/corner of the MoS₂ flake”

How does the probe do this only at the edges/corner and not at the basal sites. This is not clear and needs more of an explanation.

Figure 4 data – the basal site point, how can you convince the reader that there isn't a multilayer here? If there is even a small multi-layer component in the center of the flake then this could impact the results.

Lines 258-260 “However, the EQE and IQE values decrease significantly, consistent with lateral confinement of the electrons inside the small monolayer MoS₂ flake”. It is not clear what is causing the EQE an IQE to decrease significantly? What is this statement in regards to?

The conclusions give a perspective on bilayers leading to more higher reactivity. Can the authors comment on this from their data since some of the triangles did have bilayer formation in the middle of the triangles.

Reviewer #4

(Remarks to the Author)

The authors investigated the photochemical reactivity of 2D MoS₂ semiconductors using scanning photoelectrochemical microscopy. They proposed that photogenerated holes localize at the excitation spot, while photoelectrons exhibit high mobility and migrate away. However, similar conclusions have been validated in previous photocatalysis studies, which demonstrate that photogenerated electrons have higher mobility than holes. Additionally, while the title of the manuscript is engaging, the research content does not align with it well. Some key characterizations and discussion are missing from this article. This manuscript is recommended for rejection. Detailed comments are provided below:

1. The title of the work is “Spatially Resolved Photocatalytic Reactive Sites”, however, the research mainly validates that “oxidation products localize at the excitation spot, indicating stationary holes, while photoreduction occurs up to 80 microns away, showing exceptional electron mobility.” No experiments and discussion for identifying the specific oxidation sites and reduction sites are involved.

2. AFM measurement is suggested to perform to directly characterize the thickness of the MoS₂ nanosheets. In addition, HRTEM image should be provided to study the type of defect sites at the corner and edge of the nanosheets. Importantly, the role of these defects is suggested to be discussed.

3. In Fig. 2b, while the photogenerated holes are consumed by FcDM, what happens to the electrons? Additionally, in Fig. 2b, is hydrogen actually being produced, and if so, can it be collected?

4. The area of light irradiation is a crucial parameter for the photoactivity mapping test. Detailed information on this should be

provided for Figures 2e–f, 3a, and 4a.

5. The organization and discussion of Figures 3a and 3b are difficult to follow. It is recommended to reorganize the data to enhance clarity and make it easier for readers to understand. For example, the authors discussed that “As such, a negative (positive) photoactivity should be observed when the initial concentration of redox mediator (FcDM+) increases (decreases), accordingly to the reaction occurring at the MoS₂-liquid interface.” Did the authors change the concentration of FcDM+?

6. According to Figure 2b and c, “the probe electrochemically detects the oxidized (negative ΔI) or reduced (positive ΔI) products generated from the photocatalytic reactions occurring at the ML-MoS₂-liquid interface using different electrolytes”. However, both negative and positive current are detected in Figure 3b using FcDM electrolyte. More discussion for the reaction process or chemical equation is suggested to provide.

Version 1:

Reviewer comments:

Reviewer #1

(Remarks to the Author)

The authors' response letter provides adequate experimental supplementation and literature support. However, further clarifications are required regarding: Comparative data for innovation validation, Dynamic tracking of reaction sites, Quantitative mechanisms of multilayer/defect effects, Independent verification of H₂ production. Addressing these issues will significantly enhance the paper's rigor and academic impact. Please reference the following comments:

Questions Requiring Further Clarification and Improvement Suggestions

1. Insufficient Innovation (Reviewer 1, Comment 3)

Issue: The authors emphasize methodological innovation (e.g., regulating reaction sites via photoexcitation positioning) but lack direct comparisons with existing technologies.

Recommendations:

Add comparative experiments with electrocatalytic or traditional photocatalytic systems to demonstrate the unique advantages of photoexcitation positioning.

Include comparisons with literature-reported carrier mobility data (e.g., PL lifetime or transient absorption spectroscopy) to validate the uniqueness of long-range electron migration.

2. Dynamic Tracking of Reaction Sites (Reviewer 2, Comment 1)

Issue: The response lacks direct dynamic evidence for how photoexcitation positioning governs reaction sites.

Recommendations:

Incorporate time-resolved SPECM imaging to visualize the diffusion pathways of oxidation/reduction products after photoexcitation.

Use ultrafast spectroscopy (e.g., transient fluorescence) to study carrier migration kinetics and clarify the regulatory mechanism of photoexcitation positioning on reaction sites.

3. Quantitative Impact of Multilayer Structures (Reviewer 3, Comment 6)

Issue: Supplementary Section 3 only qualitatively discusses the effect of multilayer regions on quantum efficiency, lacking quantitative mechanistic analysis.

Recommendations:

Conduct comparative experiments on MoS₂ with varying layer numbers (monolayer, bilayer, multilayer) to establish a quantitative thickness-efficiency relationship.

Utilize scanning Kelvin probe microscopy (SKPM) or surface photovoltage (SPV) techniques to investigate the influence of multilayer regions on charge separation efficiency.

4. Direct Characterization of Defect Sites (Reviewer 4, Comment 2)

Issue: Defect types are indirectly inferred from literature without direct experimental characterization of the sample.

Recommendations:

Perform atomic-level characterization (e.g., STM or HAADF-STEM) to directly observe S-vacancy or edge defect distributions.

If experimental constraints exist, at least employ PL spectroscopy or Raman mapping (e.g., defect-related peak intensity) to indirectly assess defect density.

5. Independent Validation of H₂ Detection (Reviewer 2, Comment 2)

Issue: H₂ detection relies solely on electrochemical probes, risking interference from other reduction products.

Recommendations:

Supplement with online mass spectrometry (DEMS) or gas chromatography (GC) to independently verify H₂ generation.

Compare results under sodium thiosulfate-free conditions to exclude interference from competing reduction reactions (e.g., O₂ reduction).

6. Originality of Theoretical Calculations (Reviewer 1, Comment 2)

Issue: Theoretical analysis cites existing frameworks without customized calculations aligned with experimental findings.

Recommendations:

Perform DFT-based simulations of carrier migration pathways to explain long-range electron transfer mechanisms.

Calculate charge separation efficiency differences under varying excitation wavelengths (A/C excitons) to correlate with experimental results.

7. Title-Content Consistency (Reviewer 4, Comment 1)

Issue: The title emphasizes "atomically resolved photocatalytic active sites," yet the work fails to explicitly reveal atomic-level site information.

Recommendations:

Reviewer #2

(Remarks to the Author)

In the present manuscript, Henrotte et al. employed a combined photoelectrochemical method to study the spatial distribution of photogenerated charge carriers and their reactivities on a prototypical 2D semiconductor monolayer (ML-MoS₂). Following a thorough review of the revised manuscript, it is evident that the authors have meticulously addressed all the concerns raised, particularly the experimental details pertaining to the SPECM measurements. In light of these revisions, I believe that this version has met the requisite standards of Nature Communications and can be published without a need of further revision.

Reviewer #3

(Remarks to the Author)

The authors have added a lot of detail to their manuscript and SI which is helpful. However, I still think that more is needed to understand the difference in the photooxidation/reduction mapping to differentiate what is real and what is due to probe hinderance.

However, I am still confused by their response to the probe hinderance and how that is impacting their results. To help with this, I would suggest that the authors perform the same measurements looking at the oxidation and reduction of FcDM to convince the reader of their conclusions. By switching chemical reactions to probe reduction and oxidation, they authors add confusion about their results and the differences in photomapping.

Furthermore, I do not find the Figure S3 and S4 schematics helpful in describing the challenges with the probe hinderance and suggest they prepare different schematics to illustrate their point.

A new paper: DOI: 10.1021/acsnano.4c13276 has been published and could be helpful in the introduction to motivate this project.

Reviewer #4

(Remarks to the Author)

The revised manuscript is suggested to be published.

Version 2:

Reviewer comments:

Reviewer #1

(Remarks to the Author)

It seems OK and to be accepted. Anyways, the date is indeed insufficient. If the authors add more date, it can be awesome.

Reviewer #3

(Remarks to the Author)

The reviewers have addressed my comments and I support publishing.

Manuscript: Spatially Resolved Photocatalytic Reactive Sites and Quantum Efficiency in a 2D Semiconductor - NCOMMS-24-67854

REPLY TO THE REVIEWERS' COMMENTS

We thank the editor and the reviewers for the time spent in the revision of our manuscript. We address below point-by-point the Reviewers' comments. Please find our responses in red and highlighted in yellow the changes introduced to the manuscript and supplementary information.

Reviewer #1:

We thank the reviewer for the time spent in reviewing our manuscript. We modified our manuscript based on the reviewer's comments. We hope that the revised version can more effectively convey the significance of our findings.

Comment 1 – The number of test samples is too small, and the test results are not universal. It is suggested to increase the number of test samples and increase the supporting points of the conclusions of the paper.

Reply to Comment 1: We thank the reviewer for their comment. We added several examples in the supplementary information to support the conclusions of the paper. A new investigation was conducted for aligned-unaligned excitation-detection experiments (Supplementary Section 2) while varying the concentration of FcDM/FcDM⁺. Additionally, we repeated the experiment from Figure 4, providing an example with a smaller flake (namely Flake 2 in Fig. 4). We also present new EQE and IQE measurements, comparing cases with and without multilayer regions (Supplementary Section 3, Fig. S11 and S12). Finally, we analyzed the EQE and IQE across eight different flakes to highlight the robustness and high reproducibility of our measurements (Supplementary Section 3, Fig. S13 and S14).

The revised Fig. 4:

“Fig. 1: Spatially-resolved quantum efficiency of monolayer MoS₂. a, External quantum efficiency (*EQE*) map of monolayer (ML-) MoS₂ flakes excited at $\lambda_{660nm} = 0.52 \text{ W cm}^{-2}$ with $r_{\text{Light}} = 5 \mu\text{m}$. The dashed triangles correspond to the highlighted flakes in Fig. S5. (b) External and (c) internal quantum efficiency at different positions (Flake 1 basal: dark red; edge: grey; corner: purple; Flake 2 basal: orange; corner: light purple) inside the ML-MoS₂ according to the excited transition at $\sim 1.2 \times 10^{12} \text{ photons s}^{-1}$ (C transition at 455 nm; A transition at 660 nm). The error bars represent the standard deviation.”

The newly added Supplementary Section 2:

Line 107 in SI:

“Supplementary Section 2: Influence of the FcDM⁺ species on the observed photoactivity.

To better illustrate the effect of the FcDM⁺ species on system limitations, we performed aligned-unaligned excitation-detection measurements on freshly prepared and one-year-old 1 mM FcDM solutions. Due to the equilibrium between FcDM and FcDM⁺ in water, the aging process resulted in a shift toward a lower FcDM concentration (and a correspondingly higher FcDM⁺ concentration) in the one-year-old solution (Fig. S6).

Fig. S6: Cyclic voltammetry of 1 mM FcDM solution freshly prepared (dark) and 1-year-old (purple) representing the normalized probe current according to the probe current obtained at the oxidation peak for the 1 mM FcDM fresh solution.

This decrease in the reduced species (FcDM) and corresponding increase in the oxidized species (FcDM⁺) influence the extraction efficiency of photogenerated charge carriers during the measurements. The relative signal intensity provides direct information on species concentration, as only the species concentration varies (Eq. S7). Considering a normalized probe current of 1 for the fresh 1 mM FcDM solution, the FcDM⁺ species exhibited a normalized intensity of 0.06. In contrast, the one-year-old solution yielded normalized intensities of 0.78 and 0.15 for FcDM and FcDM⁺, respectively.

As in Fig. 3, we conducted aligned-unaligned excitation-detection measurements for both solutions and compared the results (Fig. S7). Interestingly, the aligned measurements exhibited similar photoactivities. However, the one-year-old solution displayed a slightly more negative photoactivity for both C and A transitions. The most significant difference emerged in measurements taken away from the illumination position (as indicated in Fig. S7a). While the fresh FcDM solution exhibited a signal comparable to that shown in Fig. 3, no detectable signal was observed for the one-year-old FcDM solution (Fig. S7b,c). This can be attributed to the changes in the availability of FcDM/FcDM⁺ species: the concentration of FcDM molecules available for hole extraction decreased by approximately 20%, whereas the availability of FcDM⁺ for electron extraction increased by nearly 200%. Consequently, the local concentration of FcDM⁺ became less limiting compared to the fresh FcDM solution.

Fig. S7: **a**, Optical image showing the monolayer (ML-) MoS₂ flake, the position of the light excitation spot, and the ultramicroelectrode (UME) position corresponding to the unaligned excitation-detection measurements. For the aligned excitation-detection measurements, the UME was positioned at the light excitation spot. **(b,c)** Photoactivity measurements under chopped light excitation (5 s; increased light power at every pulse) for the aligned (negative ΔI) and unaligned (positive ΔI) measurements for the C transition **(b)** and the A transition **(c)** performed in presence of fresh 1 mM FcDM (black curves) and one-year-old 1 mM FcDM (colored curves) solutions. Conditions: $r_T = 0.1 \mu\text{m}$, $R_G = 25$, and $Z = 5 \mu\text{m}$ for **(b,c)**. The initial solutions were prepared with 1 mM FcDM and 0.1 M KCl.

To further investigate the regions where reduction products formed on the MoS₂ flake immersed in the one-year-old FcDM solution, we performed unaligned excitation-detection measurements at different locations along the flake (Fig. S8). We selected four positions for probing photoproducts: (1) far from the illumination site (~75 μm), (2) mid-distance (~30 μm), (3) near the illumination site (~10 μm), and (4) directly adjacent to the illuminated region (~1–2 μm) (Fig. S8a). Notably, photoactivity associated with reduced products peaked at position 3 for the C transition (Fig. S8b) and at position 4 for the A transition (Fig. S8c). A sharp increase in signal intensity was observed immediately upon illumination, followed by a decrease, suggesting that reduction occurs instantaneously while oxidized products diffuse from the excitation site toward the probe (Fig. S8b). This effect was particularly pronounced for the C transition at position 4, where the high diffusion of FcDM⁺ resulted in a decrease in photoactivity, ultimately yielding negative ΔI values. In contrast, the species generated under excitation at the A transition exhibited little to no diffusion. This behavior may be attributed to differences in carrier dynamics, as A-excitons correspond to more localized and tightly bound carriers compared to those associated with the C transition. Furthermore, the absence of photoactivity at position 1 indicates that electrons no longer reached this region. We hypothesize that the combination of the excitation spot size and the decrease in FcDM concentration acted as limiting factors in the one-year-old FcDM solution. This highlights the critical influence of electrolyte composition (availability of species to be reduced and oxidized) and light excitation position on the photoreaction dynamics occurring at the MoS₂ platform.

Fig. S8: **a**, Optical image showing the monolayer (ML-) MoS₂ flake, the position of the light excitation spot, and the ultramicroelectrode (UME) positions corresponding to the unaligned excitation-detection measurements performed at the different positions indicated on the image: (1) far from the illumination site (~75 μm), (2) mid-distance (~30 μm), (3) near the illumination site (~10 μm), and (4) directly adjacent to the illuminated region (~1–2 μm). **(b,c)** Photoactivity measurements under chopped light excitation (5 s; increased light power at every pulse) at the different positions indicated in **(a)** for the C transition **(b)** and the A transition **(c)** performed in presence of one-year-old 1 mM FcDM solutions. Conditions: $r_T = 0.1 \mu\text{m}$, $RG = 25$, and $Z = 5 \mu\text{m}$ for **(b,c)**. The initial solution was prepared with 1 mM FcDM and 0.1 M KCl.”

The newly added Supplementary Section 3:

Line 187 in SI:

“Supplementary Section 3: Effect of the MoS₂ flake morphology on the quantum efficiency.

To investigate the influence of MoS₂ flake morphology on its photocatalytic efficiency, we selected a monolayer flake with a small multilayer region at its centre, similar to the one depicted in Fig. 1. Aligned excitation-detection measurements were performed to ensure the reliability of our results and to examine the impact of the multilayer region on quantum efficiency (QE). As shown in Fig. S11, measurements were conducted at three distinct locations: the flake's corner (purple circle), the multilayer-containing centre (dark red circle), and the basal plane, away from the centre and free of multilayer regions (grey circle).

Fig. S11: Optical image corresponding to the investigated monolayer MoS₂ flake with the different positions where measurements were performed: purple for corner, dark red for basal at the centre with the presence of multilayer (namely, basal (multi)), and grey for basal away from the centre to avoid any multilayer (namely, basal (mono)).

For each position, we evaluated the external quantum efficiency (EQE) and internal quantum efficiency (IQE) for the C and A transitions under varying photon fluxes (Fig. S12). Notably, at the basal plane, the QE exhibited a substantial difference between the C and A transitions, depending on the presence or absence of the multilayer. Specifically, the QE for the A transition was significantly reduced in the presence of a multilayer compared to a purely monolayer region (Fig. S12a,b). In contrast, the QEs for the C transition remained largely unchanged. These findings suggest that the multilayer region impedes charge carrier extraction from bound excitons at the MoS₂-liquid interface, thereby decreasing the overall QE of the system.

Interestingly, under high photon flux, this difference became less pronounced, as spatial constraints imposed by the MoS₂ flake dimensions and the availability of charge-extracting molecules in solution remained fixed (Fig. S12c,d). The increased photon flux led to a greater number of charge generation events, which enhanced charge carrier extraction efficiency until the system reached its inherent limitations.

Fig. S12: (a,c) External and (b,d) internal quantum efficiency at different positions (basal (multi): dark red; basal (mono): grey; corner: purple) inside the ML-MoS₂ (Fig. S11) according to the excited transition at (a,b) $\sim 1.2 \times 10^{12}$ and (c,d) $\sim 14 \times 10^{12}$ photons s⁻¹ (C transition at 455 nm; A transition at 660 nm). The error bars represent the standard deviation.

To assess the role of flake size in QE measurements, we studied MoS₂ flakes of varying sizes relative to the excitation beam (approximately 80 μm^2). The selected flakes ranged from 0.5 to 70 times the beam area (Fig. S13).

Fig. S13: On the left, optical images of the different flakes investigated for comparing the flake size. On the right, table containing the flake number and the corresponding area measured with ImageJ.

Measurements of EQE (Fig. S14a,c) and IQE (Fig. S14b,d) were conducted at different photon fluxes ($\sim 1.2 \times 10^{12} \text{ s}^{-1}$, Fig. S14a,b; $\sim 14 \times 10^{12} \text{ s}^{-1}$, Fig. S14c,d) at the centre of each flake, which were numbered according to their size, from the largest (Flake 1) to the smallest (Flake 8). At low photon flux, EQE and IQE remained consistent for Flakes 1 to 6 but dropped significantly for Flakes 7 and 8. This reduction was less pronounced for Flake 7 under A transition excitation. These results suggest that QE is governed by multiple factors, including flake size, excitation spot size, and the availability of charge-extracting species in solution. For flakes with an area close to the light spot size, photocarrier saturation occurred even at the lowest studied intensities. Conversely, for flakes much larger than the excitation beam (10 times or more), the maximum QE was limited by charge carrier interaction with the solution rather than by spatial confinement effects. Consequently, similar QE values were observed for all large flakes (Flakes 1 to 6).

Under higher photon flux (Fig. S14c,d), the overall trend remained unchanged, except for Flakes 5 and 6 under C transition excitation. The observed reduction in QE for these flakes suggests a transition in the limiting factor from charge carrier availability at the MoS₂-liquid interface to the MoS₂ surface area available for charge extraction.

Ultimately, our findings highlight the complexity of understanding photocatalytic behavior, even in a simple system like MoS₂. Our results underscore the importance of multiple interdependent factors in accurately assessing and comparing photocatalytic efficiency, including the availability of redox-active molecules in solution, excitation spot properties (size, wavelength, and intensity), and the MoS₂ platform characteristics (thickness, size, and defect concentration). Overall, this study presents a robust methodology for evaluating the photocatalytic properties of materials under realistic operating conditions, allowing simultaneous assessment of both reduction and oxidation reactions without measurement-induced artifacts.

Fig. S14: (a,c) External and (b,d) internal quantum efficiency of different MoS₂ flakes (Fig. S13) according to the excited transition at (a,b) $\sim 1.2 \times 10^{12}$ and (c,d) $\sim 14 \times 10^{12}$ photons s⁻¹ (C transition at 455 nm; A transition at 660 nm). The error bars represent the standard deviation. The number of the flakes was assigned from the biggest to the smallest flakes (see Fig. S13).”

Comment 2 – Lack of theoretical calculation, and lack of theoretical calculation support for photocurrent.

Reply to Comment 2: We thank the reviewer for their suggestion. There is no photocurrent recovered during our measurements. To eliminate any ambiguity, we modified the method section about the diffusion model developed with COMSOL and employed to determine the photocatalytic product collection efficiency at the UME probe. The current recorded is linked to the probe which is detecting the chemicals produced in situ. The diffusion of the species from the investigated surface to the probe unlocks the calculation of photocatalytic events occurring at the surface. We modified accordingly to avoid any misunderstanding:

Line 401: “The ML-MoS₂ local QE was determined from the chopped light measurements using a previously introduced COMSOL diffusion model simulating the substrate ΔI based on the probe ΔI .²⁸ The diffusion model was simulated using the “Transport of Diluted Species” physics in a 2D

asymmetric mode considering the substrate as an electrochemical object corresponding to the light beam size. The diffusion model provides information such as the ratio between the probe current and the substrate current (called collection efficiency, CE), the probe geometry influence and the effect of the probe-substrate distance on the CE. The external and internal quantum efficiencies were calculated as follow:

$$EQE (\%) = \frac{\Delta I_S / e}{P / h\nu}, \text{ with } CE = \frac{\Delta I_T}{\Delta I_S} \quad (\text{Eq. 3 and 4})$$

$$IQE (\%) = \frac{EQE (\%)}{\text{Absorption}} \quad (\text{Eq. 5})$$

Where ΔI_T is the measured probe current related to the photoproducts, ΔI_S corresponds to the calculated photocatalytic events effectively occurring at the surface, e is the electron charge ($1.602176634 \times 10^{-19}$ C), P is the power density of the incident light, h is the Planck constant ($6.62607015 \times 10^{-34}$ m² kg s⁻¹), ν is the photon frequency, and CE is the collection efficiency between the electrochemical probe and the investigated surface.”

Extensive theoretical calculations have already been reported on the electronic structure of MoS₂ (DOI: 10.1016/j.ssc.2012.02.005, among many others). These studies have theoretically identified the favorable pathway – depending of the phase of MoS₂ and the position (basal vs edge) – for electrocatalytic HER (DOI: 10.1021/acscatal.6b01211 and 10.1016/j.apsusc.2019.143869) and photocatalytic HER (DOI: 10.1016/j.jpcs.2024.112543). We added a sentence in the text introducing those works:

Line 146: “... Furthermore, extensive theoretical calculations have demonstrated that the electronic structure of MoS₂ for (photo)-electrocatalytic HER is highly dependent on its phase and atomic configuration.^{51–54}”

Added references:

“51. Kadantsev, E. S. & Hawrylak, P. Electronic structure of a single MoS₂ monolayer. *Solid State Communications* **152**, 909–913 (2012).

52. Tang, Q. & Jiang, D. Mechanism of Hydrogen Evolution Reaction on 1T-MoS₂ from First Principles. *ACS Catal.* **6**, 4953–4961 (2016).

53. Li, W. *et al.* Hydrogen evolution reaction mechanism on 2H-MoS₂ electrocatalyst. *Applied Surface Science* **498**, 143869 (2019).

54. Achqraoui, M., Bekkioui, N., Jebari, H. & Ez-Zahraouy, H. Exploring layer-dependent photocatalytic and optoelectronic properties of MoS₂ for hydrogen production through DFT analysis. *Journal of Physics and Chemistry of Solids* **199**, 112543 (2025).”

Comment 3 – From the perspective of innovation, although your research topic has some exploration value, unfortunately, we think that the innovation shown in the current manuscript is not outstanding enough.

Reply to Comment 3: In our study, we developed a novel methodology to investigate the impact of localized irradiation on the charge carrier population across a single MoS₂ flake. Using aligned and unaligned excitation-detection measurements, we demonstrated the pivotal role of the excitation location in governing the entire photoactivity of 2D TMDs. It is well-established that defect sites are the dominant factor in electrocatalysis. Nonetheless, our findings reveal that the reactivity in 2D TMDs photocatalysts can be precisely controlled through the strategic modulation of light excitation and its positioning, independently of the defect site locations. Specifically, we showed that the regions for oxidation and reduction reactions are dictated by the location and intensity of illumination. We have not found in the literature anything closer to this result and we think it can dramatically alter the photocatalysis field with 2D materials. We encourage the reviewer to provide such references in case we have missed them. Our insights represent a paradigm shift in how 2D TMD platforms can be designed and optimized for photocatalysis. By unveiling the interplay between excitation localization and photocatalytic activity, our findings open new avenues for the nano-engineering of TMDs, enabling tailored functionality for advanced applications.

Additionally, we have carefully revised the manuscript to address all reviewers' comments, enhancing the clarity, robustness, and relevance of our work. We believe the contributions of our study will significantly advance the understanding and practical design of 2D materials for next-generation photocatalytic systems.

Comment 4 – For Fig. 4c and Fig. S6b, the reviewer requests that the authors examine the distribution of the data points to identify any potential outliers or biases that may be contributing to the larger error bars.

Reply to Comment 4: We thank the reviewer for highlighting this point. We corrected the error bars in the revised version.

Comment 5 – The coordinates of the image are too small, which is not conducive to readers' reading. We suggest that the coordinates of the image be enlarged.

Reply to Comment 5: We thank the reviewer for his/her comment. We have improved the figures readability in the revised version.

Reviewer #2:

In this manuscript, Henrotte et al. used a SPECM-based method to study the spatial distribution of photogenerated charge carriers and their reactivities on a prototypical 2D semiconductor monolayer (ML-MoS₂). Using an interesting aligned-unaligned excitation-detection methods, the authors were able to clearly show that photooxidation is localized at the excitation spot of ML-MoS₂, while photoreduction occurs up to 80 microns away due to the exceptional mobility of the photogenerated electrons in ML-MoS₂. Overall, this work provides a valuable tool to study the photochemical reactivity of 2D TMDs, which is not yet fully understood, but several concerns also need to be addressed before publication. Therefore, I recommend this work to be published after the major revisions listed below.

We thank the reviewer for the positive evaluation of our manuscript and for highlighting our thoughtful approach in designing and realizing the aligned-unaligned excitation-detection experiments. We improved the quality of the manuscript following the reviewer suggestions/comments. The detailed replies to the reviewer's comments are reported below.

Comment 1 – Using the SPECM method, the authors showed that the reactivities of the photogenerated electrons were localized at the basal plane of ML-MoS₂ (Figure 2f). This is quite contradictory to previous views that the active sites of 2D TMDs are mainly corners and edges. In this sense, the authors failed to spatially resolve the photocatalytic reactive sites of ML-MoS₂, the main target of this study. The authors must make this point clear and explicit in the manuscript, otherwise the validity of the presented method would be compromised.

Reply to Comment 1: We thank the reviewer for bringing this important point to our attention. We acknowledge that the initial presentation of the observations in Figure 2 may have led to some misunderstanding regarding the spatial localization of the reactive sites in monolayer MoS₂. To address this, we have revised the corresponding section of the manuscript to provide a more detailed explanation of our findings, with a particular emphasis on the photoreactivity of bare MoS₂ and the implications of our observations. Although reactive sites are widely recognized as the primary contributors to electrocatalysis, our results demonstrate that the spatial reactivity of photocatalysts can be regulated by strategically modulating light excitation (i.e., location and intensity).

More importantly, we have included a dedicated section in the supplementary information that elaborates on the complexity of the SPECM (scanning photoelectrochemical microscopy) measurements. This supplementary section discusses how the methodology influences the obtained results and clarifies the relationship between the photogenerated charge carrier distribution and the observed reactivities in Fig. 2. We believe these revisions enhance the transparency and validity of our approach and ensure that its novelty is clearly articulated.

The main text was modified as follow:

Line 144: “The most abundant defects in MoS₂, sulfur vacancies and edge terminations, are well-known active centres in electrocatalytic processes such as the hydrogen evolution reaction (HER).^{15,17,18,46–50} Furthermore, extensive theoretical calculations have demonstrated that the electronic structure of MoS₂ for (photo)-electrocatalytic HER is highly dependent on its phase and atomic configuration.^{51–54} However, little is known about the role of these defects in photocatalysis under excited-state conditions, where additional factors beyond static defect chemistry come into play. The obtained results deviated from the established understanding that MoS₂ exhibits catalytic activity at defect sites, specifically at the edges, for HER. Also, the measurement method itself may influence the observed activity as the proximity of the probe to the surface could introduce a hindering effect on the diffusion of the species from the MoS₂ to the solution.^{55”}

Taking into consideration the influence of the probe during our measurements (i.e., impeding the mass transport of the produced species),⁵⁵ we suggest that the probe itself hindered the diffusion of the generated products from the MoS₂ flake (see Supplementary Section 1). This hindering effect decreased the apparent photoactivity at the basal plane for the detection of oxidized products (Fig. 2e, Fig. S3), while it increased the concentration of reduced products during the photoreduction map at the same location (Fig. 2f, Fig. S4).

To verify the distribution of the reactive sites along the MoS₂ flake, we performed spatially-resolved investigations with aligned and unaligned excitation/detection positions to determine the underlying mechanism behind our observations (Fig. 2e,f). The measurements were performed with a nanoprobe at a distance ($Z = 5 \mu\text{m}$) larger than two times the radius of the probe insulating part (r_{ins}) avoiding the hindering effect of the probe on the diffusion of species. The FcDM/FcDM⁺ redox couple was used to decipher the redox reaction occurring at the probe position. Thus, the probe was positioned away from the excitation spot (Fig. 3a) or, alternatively, aligned with it (Fig. 3b). The oxidation (reduction) process occurs when the photogenerated hole (electron) is extracted from the MoS₂ by the surrounding molecules at the MoS₂-liquid interface. As such, a negative (positive) photoactivity should be observed when the initial concentration (C_{Dark}) of redox mediator (FcDM⁺) increases (decreases), accordingly to the reaction occurring at the MoS₂-liquid interface under illumination (C_{Light}).

Interestingly, the measured photoactivity was positive (reduction) when the probe was away from the excitation spot (Fig. 3c), but always negative (oxidation) when the probe was aligned with the light spot (Fig. 3d). In both cases, a rapid steady-state was observed revealing no influence from diffusion of the species.⁵⁶ This shows that the photogenerated charge carriers were extracted at the reactive sites in accordance with the observed redox process: holes at the light excitation spot, and electrons away from the light excitation spot (Fig. 3e). Since the sign of ΔI remained consistent across all measurements at a single probe position, this suggests that, regardless of the excited optical transitions, the holes are relatively static, while the electrons exhibit significantly higher mobility, traveling through the ML-MoS₂. These results provide insight into our previous observations for the aligned measurements, which only detected the products from photo-oxidation (Fig. 2e) or photoreduction (Fig. 2f) diffusing to the probe. This confirmed a significant charge carrier separation, where hole extraction is relatively localized at the excitation area, and electrons

exhibit high mobility throughout the ML-MoS₂ (Fig S4b). Furthermore, this corroborates our discussions on the hindrance of the probe on the diffusion of photoproducts from the MoS₂ to the probe (Supplementary Section 1), reinforcing the spatially selective nature of the observed photo-induced processes (Fig. 3e).”

Added references:

46. Hong, J. et al. Exploring atomic defects in molybdenum disulphide monolayers. *Nat Commun* 6, 6293 (2015).

47. Vancsó, P. et al. The intrinsic defect structure of exfoliated MoS₂ single layers revealed by Scanning Tunneling Microscopy. *Sci Rep* 6, 29726 (2016).

48. Chow, P. K. et al. Defect-Induced Photoluminescence in Monolayer Semiconducting Transition Metal Dichalcogenides. *ACS Nano* 9, 1520–1527 (2015).

49. Bretscher, H. et al. Rational Passivation of Sulfur Vacancy Defects in Two-Dimensional Transition Metal Dichalcogenides. *ACS Nano* 15, 8780–8789 (2021).

50. Li, L. et al. Role of Sulfur Vacancies and Undercoordinated Mo Regions in MoS₂ Nanosheets toward the Evolution of Hydrogen. *ACS Nano* 13, 6824–6834 (2019).

51. Kadantsev, E. S. & Hawrylak, P. Electronic structure of a single MoS₂ monolayer. *Solid State Communications* 152, 909–913 (2012).

52. Tang, Q. & Jiang, D. Mechanism of Hydrogen Evolution Reaction on 1T-MoS₂ from First Principles. *ACS Catal.* 6, 4953–4961 (2016).

53. Li, W. et al. Hydrogen evolution reaction mechanism on 2H-MoS₂ electrocatalyst. *Applied Surface Science* 498, 143869 (2019).

54. Achqraoui, M., Bekkioui, N., Jebari, H. & Ez-Zahraouy, H. Exploring layer-dependent photocatalytic and optoelectronic properties of MoS₂ for hydrogen production through DFT analysis. *Journal of Physics and Chemistry of Solids* 199, 112543 (2025).”

We also explained in more detail this hindrance effect in the new Supplementary Section 1.

Line 44, in the new Supplementary Section 1, for the photo-oxidation map:

“The electrochemical probe was biased at a potential corresponding to the diffusion limiting current of the species of interest (-0.1 V vs Ag/AgCl, KCl 3.4M), corresponding to FcDM⁺ reduction in this case. Consequently, the measured photoactivity ($\Delta I = I_{T,Light} - I_{T,Dark}$) reflects the number of FcDM molecules effectively oxidized by the MoS₂ and diffusing to the probe. Due to the counter reaction occurring at the MoS₂ flake and the hindrance caused by the probe, the

apparent activity of the basal plane (Fig. S3a) and the corner (Fig. S3b) are influenced by the surrounding species produced away from the excitation spot. Therefore, the measured $|\Delta I|$ is decreased due to photoreduction occurring around the excitation spot and the diffusion of species being impeded by the probe.

Fig. S3: Schemes describing the situation where the probe is detecting the species at the basal plane (a) or at the edge (b) of the MoS₂ flake during the photo-oxidation map. The probe size, the light illumination spot, and the distance between the probe and the investigated material represent the conditions during to the photo-oxidation map presented in Fig. 2e, according to the scale bar of 5 μm .”

Line 76, in the new Supplementary Section 1, for the photoreduction map and related hindering effect:

“In the case where the highest photoactivity was observed at the basal plane in Fig. 2f, two possible scenarios can explain our results: (1) H₂ is produced directly at the light excitation spot and subsequently detected by the probe (Fig. S4a), or (2) H₂ is produced at reactive sites and diffuses to the probe due to mass transport hindrance caused by the probe itself (Fig. S4b). Considering that the reactive sites of MoS₂ are well-known to be defect sites, as evidenced by PL mapping at the edges of the MoS₂ (Fig. 1a), the second scenario is the most plausible explanation for our observations.

To understand the hindering effect from the probe on the measurements, a dimensionless parameter has been previously introduced as L ,⁵ corresponding to the ratio between the probe-substrate distance (Z) and the active part radius (r_T). For $L \geq 10$, no hindering effect is expected. For the measurements in Fig. 2f, $L = 1$, which suggests a significant hindrance from the system on the diffusion, forcing the molecules to diffuse from the MoS₂ to the probe (Fig. S4b). Furthermore, we believe that the detection of H₂ in this system was possible due to the probe hindrance, as bare MoS₂ exhibits significantly low photogeneration of H₂.

To avoid this contribution in the measurements performed in Fig. 3 and 4, a nanoprobe ($r_T = 100$ nm) at a probe-substrate distance of 5 μm was used, corresponding to $L = 50$. Moreover, Z was chosen to keep a distance larger than $2 r_{ins}$ to completely remove any suspicions coming from the hindering effect of the nanoprobe during those measurements.

Fig. S4: Schemes describing the proposed scenarios, where the probe detects the species generated directly at the basal plane due to the photoreduction process occurring locally at the light excitation (a), or the probe detects the species generated at the edge of the MoS₂ flake due to the electron migration to the reactive sites and the subsequent hindered diffusion of the species related to the probe size (b). The probe size, the light illumination spot, and the distance between the probe and the investigated material represent the conditions during to the photoreduction map presented in Fig. 2f, according to the scalebar of 5 μm.”

Comment 2 – The authors stated that they directly measured the spatial reactivity of photogenerated electrons through electrochemically detecting the H₂ products. However, this photochemical study was only carried out without external electrochemical bias or Pt cocatalysts, which means that H₂ production might be relatively difficult. It appears that no solid evidence has been provided for the formation of H₂ on an ML-MoS₂ flask and they unambiguously measured the H₂ products rather than other reductive substances in this study. The authors might need to further clarify this point.

Reply to Comment 2: We thank the reviewer for pointing out this critical aspect in our manuscript. The reviewer is right that the H₂ photogeneration is a difficult process to occur on bare MoS₂, as the low measured probe current reveals. The H₂ generation is directly detected above the investigated surface with a platinum probe, which is biased at a potential to oxidize the generated H₂ (0.1 V vs Ag/AgCl). In our conditions, the ORR would start at a lower potential (< -0.1 V). This ensures that H₂ oxidation is the sole contribution increasing the probe current during our measurements.

As reference, the diffusion limited current of the probe ($r_T = 5 \mu\text{m}$) would be $\sim 1.2 \text{ nA}$ for the oxidation of 1 mM of FcDM; if we consider a probe current of 0.5 pA, the diffusion of H₂ in water around $4.5 \times 10^{-5} \text{ cm}^2/\text{s}$ (DOI: 10.1021/acs.jced.3c00085), the faraday constant and the active radius of $5 \times 10^{-4} \text{ cm}$, the H₂ concentration detected at the probe corresponded to $\sim 28.8 \text{ nM}$. We added this information in the Supplementary Section 1.

We modified the text and caption as follow:

Line 131: “In this modality, the probe, biased at a potential to selectively collect the chemical of interest,⁴⁵ electrochemically detects the oxidized (negative ΔI) or reduced (positive ΔI) products generated from the photocatalytic reactions occurring at the ML-MoS₂-liquid interface (Fig. 2b,c).”

The added reference:

“45. Askarova, G. *et al.* Photo-scanning Electrochemical Microscopy Observation of Overall Water Splitting at a Single Aluminum-Doped Strontium Titanium Oxide Microcrystal. *J. Am. Chem. Soc.* **145**, 6526–6534 (2023).”

We modified the caption of Fig 2.: “**Fig. 2: Photocatalytic maps of oxidation and reduction products for monolayer MoS₂.** **a**, Scheme depicting the aligned SPECAM measurements performed on ML-MoS₂ for the photo-oxidation reaction. **(b),(c)** Illustrations describing the substrate generation – tip collection mode employed in this study for detecting the photocatalytic **(b)** oxidation and **(c)** reduction products. **d**, Optical image of ML-MoS₂ deposited on SiO₂ with the dashed red square corresponding to the investigated area in **(e)** and **(f)**. **(e),(f)** Photoactivity maps of a ML-MoS₂ (black dashed triangle) corresponding to the highlighted flake in **(d)** for oxidation **(e)** and reduction **(f)** products. Conditions: $r_T = 5 \mu\text{m}$, $RG = 50$, $v_{scan} = 5 \mu\text{m s}^{-1}$, with $E_T = -0.1 \text{ V vs Ag/AgCl}$ and $Z = 20 \mu\text{m}$ for **(e)**, and $E_T = 0.1 \text{ V vs Ag/AgCl}$ and $Z = 5 \mu\text{m}$ for **(f)**. The experiments were performed in aqueous solutions containing 1 mM FcDM and 0.1 M KCl **(e)** and 0.1 M Na₂SO₄ and 10 mM Na₂SO₃ **(f)**.”

And, we added the following paragraph in the Supplementary Section 1:

Line 66 “From the ΔI ($\sim 0.5 \text{ pA}$), we can estimate the concentration of H₂ collected at the probe:

$$I_T = 4nFCDr_T$$

where n is the number of electrons involved (2); F , the faraday constant (96485.33 s.A/mol); C , the concentration in Red/Ox species; and D , the diffusion coefficient of the Red/Ox species ($4.5 \times 10^{-5} \text{ cm}^2/\text{s}$ for H₂).⁴ Thus, a concentration of 28.8 nM of H₂ can be calculated from the photoactivity recorded at the probe, revealing a poor reactivity if we consider the incident light of $7.64 \text{ W}\cdot\text{cm}^{-2}$. For comparison, the FcDM oxidation estimated concentration detected at the probe (corresponding to -7 pA) is $4.6 \mu\text{M}$. Consequently, the H₂ generation is at least 80 times less effective than the FcDM oxidation despite an incident illumination 20 times higher and a probe-distance 4 times lower.”

Comment 3, part 1 – Regarding the SPECAM method, did the authors strictly remove dissolved oxygen? In Figure 2b, what is the reduction product driven by the photogenerated electrons? Whether the photochemical reduction product would affect the SPECAM detection of FcDM+?

Reply to Comment 3, part 1: We thank the reviewer for this important comment. The measurements were performed under ambient atmosphere. The photoreduction product from the experiments in FcDM is FcDM itself from the FcDM⁺ species. As evidenced by Fig. S3, there is always the presence of FcDM⁺ in solution due to the equilibrium in water between FcDM/FcDM⁺. We added the counter reaction on the schematic to avoid any misunderstanding (Fig. 2b).

If the charge carriers were locally in similar population, we would record a photoactivity of 0 pA due to the counter-reaction happening at the same position. Furthermore, the probe potential is higher than the potential expected to detect oxygen through ORR at the platinum probe in our

conditions (10.1002/celc.202100710). Additionally, we provided below an example of aligned-unaligned excitation-detection measurements done at the diffusion limited plateau for the oxidation process (and so recording the amount of FcDM instead of FcDM⁺) at 0.4 V vs Ag/AgCl (Fig. R1). All the other conditions of the measurement were exactly the same as in Fig. 3 in the main manuscript. This provides strong evidence that the measured current is related to FcDM and FcDM⁺ species.

Figure R1: The figure represents a similar experiment than the one in Fig. 3. Tip current (I_{Tip}) measured far from the light excitation spot (> 27 pA), and aligned with the light excitation spot (< 27 pA), while the ML-MoS₂ was excited under chopped light (5 s; increased light power at every pulse; transition C, 455 nm, purple; transition B, 595 nm, yellow; transition A, 660 nm, dark red).

Comment 3, part 2 – Another question concerns the use of sulfite (SO₃²⁻) in measuring the photoreduction activities of ML-MoS₂ (Figure 2c). In this case, the authors used SO₃²⁻ to remove the photogenerated holes. Since the by-products, sulfite radicals, were well-known strong oxidants, this might destroy the structure of ML-MoS₂ and also affect the detection of the photoreduction activities of ML-MoS₂ (e.g., the consumption of photogenerated electrons by this radical would result in a much lower production yield of H₂). The authors are recommended to use other sacrificial reagents instead of sulfite anions, or check the influences of different sacrificial reagents.

Reply to Comment 3, part 2: We thank the reviewer for their remark. We agree with the reviewer that sulfite radicals are strong oxidants. However, the typical reaction occurring in water under ambient conditions is the formation of sulfate anions such as: $\text{SO}_3^{2-} + 2\text{h}^+ + \text{H}_2\text{O} \rightarrow \text{SO}_4^{2-} + 2\text{H}^+$ (DOI: 10.1016/1010-6030(94)03977-3). In our previous scheme, we showed only the first elementary step, without further consideration. To avoid any misunderstanding, we modified accordingly the schematic and added the corresponding reactions in the Supplementary Section 1.

Moreover, MoS₂ was already presented as a great and stable catalyst to perform water depollution through sulfite activation (DOI: 10.1016/j.apcatb.2018.11.061).

Additionally, we provided different map for the H₂ generation (light on probe current maps for different wavelength, Fig. R2.b-d; and a dark probe current map, Fig. R2.a), showing the stability and reliability of the MoS₂ flakes during our measurements. A slight shift in the dark current is observed due to the solution evolution over time.

Figure R2. Light off (a) and light on (b) probe current maps corresponding to the photoactivity map presented in Fig. 2f in the main text. (c) Light on (at 595 nm, 7.64 W.cm⁻²) probe current map performed in the same conditions than (b). (d) Light on (at 660 nm, 6.45 W.cm⁻²) probe current map performed in the same conditions outside of the incident light. Conditions: $r_T = 5 \mu\text{m}$, $RG = 50$, $v_{scan} = 5 \mu\text{m s}^{-1}$, $r_{Light} = 5 \mu\text{m}$, with $E_T = 0.1 \text{ V vs Ag/AgCl}$ and $Z = 5 \mu\text{m}$.

The revised Figure 2 is the following:

“Fig. 3: Photocatalytic maps of oxidation and reduction products for monolayer MoS₂. **a**, Scheme depicting the aligned SPECIM measurements performed on ML-MoS₂ for the photo-oxidation reaction. **(b),(c)** Illustrations describing the substrate generation – tip collection mode employed in this study for detecting the photocatalytic **(b)** oxidation and **(c)** reduction products. **d**, Optical image of ML-MoS₂ deposited on SiO₂ with the dashed red square corresponding to the investigated area in **(e)** and **(f)**. **(e),(f)** Photoactivity maps of a ML-MoS₂ (black dashed triangle) corresponding to the highlighted flake in **(d)** for oxidation **(e)** and reduction **(f)** products. Conditions: $r_T = 5 \mu\text{m}$, $RG = 50$, $v_{scan} = 5 \mu\text{m s}^{-1}$, $r_{Light} = 5 \mu\text{m}$, with $E_T = -0.1 \text{ V vs Ag/AgCl}$ and $Z = 20 \mu\text{m}$ for **(e)**, and $E_T = 0.1 \text{ V vs Ag/AgCl}$ and $Z = 5 \mu\text{m}$ for **(f)**. The experiments were performed in aqueous solutions containing 1 mM FcDM and 0.1 M KCl **(e)** and 0.1 M Na₂SO₄ and 10 mM Na₂SO₃ **(f)**.”

The added section in Supplementary Section 1 is the following:

“For the photoreduction experiments, water electroreduction is the most studied catalytic reaction on MoS₂. As such, we investigated the water reduction under photocatalytic conditions on bare MoS₂ monolayer, which is usually performed with hybrid materials.^{1,2} We performed the measurements in the presence of a hole scavenger to enhance the photogenerated electron lifetime. The following reactions occurred during photoreduction experiments:³

Comment 4 – The aligned-unaligned excitation-detection method is an interesting experiment. My concern is why the authors only use the FcDM-KCl system (for the determination of hole activity) to detect the reactivities. In Figure 3b, which substance is oxidized with respect to the measured positive current? Is it necessary to perform both FcDM-KCl and Na₂SO₃-Na₂SO₄ measurements for each aligned or unaligned experiment? This will help readers better understand these experiments.

Reply to Comment 4: We thank the reviewer for this suggestion. We improved the clarity of this part by adding a scheme showing the FcDM/FcDM⁺ presence in solution and reaction at the MoS₂ and the Pt probe. The experiment in presence of Na₂SO₃/Na₂SO₄ served to highlight the possibility to generate H₂ photocatalytically, and bring the observation of the H₂ detected at the basal plane. This observation led us to measure the photoreactivity through aligned and unaligned measurements, which supported our explanations according to the hindrance effect of the probe on the diffusion of the species. The oxidation products were detected above the light excitation, while the reduction products were detected away from the light excitation. We hope that the revised version explains better this point and improved the clarity of our findings. The manuscript was modified accordingly.

Revised Fig. 3 with the additional schemes for clarity:

“Fig. 4: Aligned-unaligned excitation-detection experiments in monolayer MoS₂. (a),(b) Illustrations depicting the measurements performed with the nanoprobe when the excitation – detection was unaligned (a), or aligned (b). (c),(d) Photoactivity (ΔI) measured far from the light excitation spot (c), and aligned with the light excitation spot (d), while the ML-MoS₂ was excited under chopped light (5 s; increased light power at every pulse; transition C, 455 nm, purple; exciton B, 595 nm, yellow; exciton A, 660 nm, dark red). The probe positions correspond to the respective

inset image, showing the monolayer (ML-) MoS₂ flake (white dashed triangle), the position of the light excitation spot (black dashed circle), and the ultramicroelectrode (UME) position (red dashed circle). **e**, Scheme representing the spatial distribution of the photo-redox processes detected at the UME in ML-MoS₂ when excited at a fixed position, with the corresponding photogenerated charge carriers (oxidation: holes, blue region; reduction: electrons, red region). Conditions: $r_T = 0.1 \mu\text{m}$, $RG = 25$, $r_{\text{Light}} = 5 \mu\text{m}$ and $Z = 5 \mu\text{m}$ for **(d)**,**(e)**. The experiments were performed in aqueous solutions containing 1 mM FcDM and 0.1 M KCl. FcDM⁺ is present in low concentration due to the redox couple equilibrium in water (Fig. S5).”

Comment 5 – The authors emphasized the numerous advantages of SPECM over other surface photovoltage and fluorescence microscopies (Line 73-77). It is recommended to cite several important papers to help readers better understand this work, for example, Nature, 2022, 610, 296–301; Nature, 2016, 530, 77–80; Sci. Adv. 2021, 7, eabj4452; J. Phys. Chem. Lett. 2023, 14, 5410–5425.

Reply to Comment 5: We thank the reviewer for this comment. We have now rewritten this part, citing and emphasizing the contributions of several key papers in the field (including the ones suggested), outlined the shortcomings of the techniques, and explained how SPECM can be advantageous in this respect.

Line 74: “... chemical reactivity. Recent imaging techniques, such as surface photovoltage and fluorescence microscopy, have provided plenty of valuable insights into the microscopic excited-state dynamics and photochemical reactivity of a variety of catalysts.^{22–29} However, these methods offer extreme spatial resolution but lack in chemical specificity. In contrast, scanning photoelectrochemical microscopy (SPECM) enables direct quantum efficiency mapping with high spatial resolution (~200 nm) and allows local quantification of redox reactions (e.g., H₂ evolution) under light excitation, providing a more direct assessment of catalytic performance.”

Added references:

22. Sambur, J. B. et al. Sub-particle reaction and photocurrent mapping to optimize catalyst-modified photoanodes. Nature 530, 77–80 (2016).

23. Mao, X. & Chen, P. Inter-facet junction effects on particulate photoelectrodes. Nat. Mater. 21, 331–337 (2022).

24. Lv, M., Zhang, X., Li, B., Huang, B. & Zheng, Z. Single-Particle Fluorescence Spectroscopy for Elucidating Charge Transfer and Catalytic Mechanisms on Nanophotocatalysts. ACS Nano 18, 30247–30268 (2024).

25. Li, R. et al. Spatial separation of photogenerated electrons and holes among {010} and {110} crystal facets of BiVO₄. Nat Commun 4, 1432 (2013).

26. Huang, T.-X. et al. Single-molecule photocatalytic dynamics at individual defects in two-dimensional layered materials. *Science Advances* 7, eabj4452 (2021).
27. Chen, R. et al. Spatiotemporal imaging of charge transfer in photocatalyst particles. *Nature* 610, 296–301 (2022).
28. Chen, R. et al. Charge separation via asymmetric illumination in photocatalytic Cu₂O particles. *Nat Energy* 3, 655–663 (2018).
29. Zhang, Y. et al. Superresolution fluorescence mapping of single-nanoparticle catalysts reveals spatiotemporal variations in surface reactivity. *Proceedings of the National Academy of Sciences* 112, 8959–8964 (2015).”

Reviewer #3:

This paper by Henrotte et al. titled “Spatially resolved photocatalytic reactive sites and quantum efficiency in a 2D semiconductor” details HER at basal vs corner sites and as a function of photoexcitation distance. This is very exciting work by the authors. I would recommend publication after minor revisions.

We thank the reviewer for the very positive assessment of our work. We improved the manuscript following the reviewer’s comments/suggestions. The detailed replies to the reviewer’s comments are reported below.

Comment 1 – Abstract: “Oxidation products localize at the excitation spot, indicating stationary holes, while photoreduction occurs up to 80 microns away, showing exceptional electron mobility”. This distance is limited by the size of the triangle shown in the SI. This should be more clearly stated that the distance is limited by the size of the triangle and that is the upper limit in these sets of experiments. Also can the authors suggest from their data how far they think the electron could travel if it the triangle was larger.

Reply to Comment 1: We thank the reviewer for their remark. We added a sentence explaining the limitation related to the size of our MoS₂ flakes. Considering the maximum distance expected, this distance is influenced by the quality of MoS₂. Nevertheless, we have preliminary results showing transport of charge carriers hundreds of microns away in connected flakes.

We modified the text as follow:

Abstract: “Oxidation products localize at the excitation spot, indicating stationary holes, while photoreduction occurs up to **at least** 80 microns away, showing exceptional electron mobility.”

Line 238: “The photoactivity evolution, observed even 80 μm away from the excitation spot, suggests a remarkable mobility of electrons inside the ML-MoS₂ under continuous wave illumination, in light of the previously reported diffusion lengths of excitons under mild conditions in PL measurements (< 2 μm).^{35–38} **Notably, the size of the ML-MoS₂ flake was the key parameter limiting the observed diffusion length in our experiments.**”

Comment 2 – “We also elucidate the photochemical reactivity according to the nature of the electronic excitation, showing that strongly-bound A-excitons outperform weakly-bound (free-carrier like) C-excitons across the flake.” This is true for IQE and not EQE. This should be specifically said in the abstract to not mislead the reader.

Reply to Comment 2: We thank the reviewer for his/her observation. We corrected the abstract accordingly.

Abstract:

“We also elucidate the photochemical reactivity according to the nature of the electronic excitation, showing that **the internal quantum efficiency of** strongly-bound A-excitons outperform weakly-bound (free-carrier like) C-excitons across the flake.”

Comment 3 – Figure 2b what is the reduction reaction to balance the oxidation reaction?

Reply to Comment 3: We thank the reviewer for this question. The reduction reaction is the FcDM^+ reduction. We added the counter reaction on the schematic in Fig. 2b.

Revised Fig. 2 with the correct reactions occurring in b and c:

“Fig. 5: Photocatalytic maps of oxidation and reduction products for monolayer MoS₂. a, Scheme depicting the aligned SPECM measurements performed on ML-MoS₂ for the photo-oxidation reaction. **(b),(c)** Illustrations describing the substrate generation – tip collection mode employed in this study for detecting the photocatalytic **(b)** oxidation and **(c)** reduction products. **d,** Optical image of ML-MoS₂ deposited on SiO₂ with the dashed red square corresponding to the investigated area in **(e)** and **(f)**. **(e),(f)** Photoactivity maps of a ML-MoS₂ (black dashed triangle) corresponding to the highlighted flake in **(d)** for oxidation **(e)** and reduction **(f)** products. Conditions: $r_T = 5 \mu\text{m}$, $RG = 50$, $v_{scan} = 5 \mu\text{m s}^{-1}$, $r_{Light} = 5 \mu\text{m}$, with $E_T = -0.1 \text{ V vs Ag/AgCl}$ and $Z = 20 \mu\text{m}$ for **(e)**, and $E_T = 0.1 \text{ V vs Ag/AgCl}$ and $Z = 5 \mu\text{m}$ for **(f)**. The experiments were performed in aqueous solutions containing 1 mM FcDM and 0.1 M KCl **(e)** and 0.1 M Na₂SO₄ and 10 mM Na₂SO₃ **(f)**.”

Comment 4 & 5 – Figure 2 e and f – why are the oxidation products (holes) more favorable on the edges? I would have thought that the edges contain more sulfur vacancies and would trap

electrons? It would be good to understand why the holes are trapped at the edges/or alternatively why the electrons stay at the basal sites.

Line 162 – 167 “Taking into consideration the influence of the probe.... The probe itself hindered the diffusion of the generated products at the edge/corner of the MoS₂ flake”. How does the probe do this only at the edges/corner and not at the basal sites. This is not clear and needs more of an explanation.

Reply to Comment 4 & 5: We thank the reviewer for pointing out the lack of clarity. We do not think that the holes are trapped at the edges and electrons stay at the basal sites. The light excitation spot is the deciding factor for the charge carrier population along the MoS₂ flake. The probe hindrance is the reason for the observations in Figure 2 e and f. We added a schematic of the probe influence and the possible scenarios connected to this to simplify the understanding of our manuscript in the new Supplementary Section 1. We also modified the main manuscript accordingly to improve the clarity of our observations.

Line 148: “However, little is known about the role of these defects in photocatalysis under excited-state conditions, where additional factors beyond static defect chemistry come into play. The obtained results deviated from the established understanding that MoS₂ exhibits catalytic activity at defect sites, specifically at the edges, for HER. Also, the measurement method itself may influence the observed activity as the proximity of the probe to the surface could introduce a hindering effect on the diffusion of the species from the MoS₂ to the solution.⁵⁵”

Taking into consideration the influence of the probe during our measurements (i.e., impeding the mass transport of the produced species),⁵⁵ we suggest that the probe itself hindered the diffusion of the generated products from the MoS₂ flake (see Supplementary Section 1). This hindering effect decreased the apparent photoactivity at the basal plane for the detection of oxidized products (Fig. 2e, Fig. S3), while it increased the concentration of reduced products during the photoreduction map at the same location (Fig. 2f, Fig. S4).

To verify the distribution of the reactive sites along the MoS₂ flake, we performed spatially-resolved investigations with aligned and unaligned excitation/detection positions to determine the underlying mechanism behind our observations (Fig. 2e,f). The measurements were performed with a nanoprobe at a distance ($Z = 5 \mu\text{m}$) larger than two times the radius of the probe insulating part (r_{ins}) avoiding the hindering effect of the probe on the diffusion of species. The FcDM/FcDM⁺ redox couple was used to decipher the redox reaction occurring at the probe position. Thus, the probe was positioned away from the excitation spot (Fig. 3a) or, alternatively, aligned with it (Fig. 3b). The oxidation (reduction) process occurs when the photogenerated hole (electron) is extracted from the MoS₂ by the surrounding molecules at the MoS₂-liquid interface. As such, a negative (positive) photoactivity should be observed when the initial concentration (C_{Dark}) of redox mediator (FcDM⁺) increases (decreases), accordingly to the reaction occurring at the MoS₂-liquid interface under illumination (C_{Light}).

Interestingly, the measured photoactivity was positive (reduction) when the probe was away from the excitation spot (Fig. 3c), but always negative (oxidation) when the probe was aligned with the light spot (Fig. 3d). In both cases, a rapid steady-state was observed revealing no influence from diffusion of the species.⁵⁶ This shows that the photogenerated charge carriers were extracted at the reactive sites in accordance with the observed redox process: holes at the light excitation spot, and electrons away from the light excitation spot (Fig. 3e). Since the sign of ΔI remained consistent across all measurements at a single probe position, this suggests that, regardless of the excited optical transitions, the holes are relatively static, while the electrons exhibit significantly higher mobility, traveling through the ML-MoS2. These results provide insight into our previous observations for the aligned measurements, which only detected the products from photo-oxidation (Fig. 2e) or photoreduction (Fig. 2f) diffusing to the probe. This confirmed a significant charge carrier separation, where hole extraction is relatively localized at the excitation area, and electrons exhibit high mobility throughout the ML-MoS2 (Fig S4b). Furthermore, this corroborates our discussions on the hindrance of the probe on the diffusion of photoproducts from the MoS2 to the probe (Supplementary Section 1), reinforcing the spatially selective nature of the observed photoprocesses (Fig. 3e).”

We also explained in more detail this hindrance effect in the new Supplementary Section 1.

Line 44, in the new Supplementary Section 1, for the photo-oxidation map:

“The electrochemical probe was biased at a potential corresponding to the diffusion limiting current of the species of interest (-0.1 V vs Ag/AgCl, KCl 3.4M), corresponding to FcDM⁺ reduction in this case. Consequently, the measured photoactivity ($\Delta I = I_{T,Light} - I_{T,Dark}$) reflects the FcDM molecules effectively oxidized by the MoS2 and diffusing to the probe. Due to the counter reaction occurring at the MoS2 flake and the hindrance caused by the probe, the apparent activity of the basal plane (Fig. S3a) and the corner (Fig. S3b) are influenced by the surrounding species produced away from the excitation spot. Therefore, the measured $|\Delta I|$ is decreased due to photoreduction occurring around the excitation spot and the diffusion of species being impeded by the probe.

Fig. S3: Schemes describing the situation where the probe is detecting the species at the basal plane (a) or at the edge (b) of the MoS2 flake during the photo-oxidation map. The probe size, the

light illumination spot, and the distance between the probe and the investigated material represent the conditions during to the photo-oxidation map presented in Fig. 2e, according to the scale bar of 5 μm .”

Line 76, in the new Supplementary Section 1, for the photoreduction map and related hindering effect:

“In the case where the highest photoactivity was observed at the basal plane in Fig. 2f, two possible scenarios can explain our results: (1) H_2 is produced directly at the light excitation spot and subsequently detected by the probe (Fig. S4a), or (2) H_2 is produced at reactive sites and diffuses to the probe due to mass transport hindrance caused by the probe itself (Fig. S4b). Considering that the reactive sites of MoS_2 are well-known to be defect sites, as evidenced by PL mapping at the edges of the MoS_2 (Fig. 1a), the second scenario is the most plausible explanation for our observations.

To understand the hindering effect from the probe on the measurements, a dimensionless parameter has been previously introduced as L ,⁵ corresponding to the ratio between the probe-substrate distance (Z) and the active part radius (r_T). For $L \geq 10$, no hindering effect is expected. For the measurements in Fig. 2f, $L = 1$, which suggests a significant hindrance from the system on the diffusion, forcing the molecules to diffuse from the MoS_2 to the probe (Fig. S4b). Furthermore, we believe that the detection of H_2 in this system was possible due to the probe hindrance, as bare MoS_2 exhibits significantly low photogeneration of H_2 .

To avoid this contribution in the measurements performed in Fig. 3 and 4, a nanoprobe ($r_T = 100$ nm) at a probe-substrate distance of 5 μm was used, corresponding to $L = 50$. Moreover, Z was chosen to keep a distance larger than $2 r_{\text{ins}}$ to completely remove any suspicions coming from the hindering effect of the nanoprobe during those measurements.

Fig. S4: Schemes describing the proposed scenarios, where the probe detects the species generated directly at the basal plane due to the photoreduction process occurring locally at the light excitation spot (a), or the probe detects the species generated at the edge of the MoS_2 flake due to the electron migration to the reactive sites and the subsequent hindered diffusion of the species related to the probe size (b). The probe size, the light illumination spot, and the distance between the probe and the investigated material represent the conditions during to the photoreduction map presented in Fig. 2f, according to the scalebar of 5 μm .”

Comment 6 – Figure 4 data – the basal site point, how can you convince the reader that there isn't a multilayer here? If there is even a small multi-layer component in the center of the flake then this could impact the results.

Reply to Comment 6: We thank the reviewer for this critical remark. The reviewer is right, there is a multilayer at the centre of the MoS₂ flake and can influence the obtained results. To understand better the effect of this multilayer and avoid any misinterpretation surrounding our previous interpretation, we added several examples of MoS₂ flakes to explore the influence of this multilayer on the obtained quantum efficiencies (QEs). We redid the experiment on the Flake 1 (Fig. 4), redoing basal and corner, as well as an edge position. We also added the QEs for Flake 2 (Fig. 4). Furthermore, we dedicated a new Supplementary Section (see Supplementary Section 3) highlighting different examples of MoS₂ flakes to be as exhaustive and transparent as we could on the effect of the MoS₂ flake morphology.

The newly added Supplementary Section 3:

Line 187 in SI:

“Supplementary Section 3: Effect of the MoS₂ flake morphology on the quantum efficiency.

To investigate the influence of MoS₂ flake morphology on its photocatalytic efficiency, we selected a monolayer flake with a small multilayer region at its centre, similar to the one depicted in Fig. 1. Aligned excitation-detection measurements were performed to ensure the reliability of our results and to examine the impact of the multilayer region on quantum efficiency (QE). As shown in Fig. S11, measurements were conducted at three distinct locations: the flake's corner (purple circle), the multilayer-containing centre (dark red circle), and the basal plane, away from the centre and free of multilayer regions (grey circle).

Fig. S11: Optical image corresponding to the investigated monolayer MoS₂ flake with the different positions where measurements were performed: purple for corner, dark red for basal at the centre with the presence of multilayer (namely, basal (multi)), and grey for basal away from the centre to avoid any multilayer (namely, basal (mono)).

For each position, we evaluated the external quantum efficiency (EQE) and internal quantum efficiency (IQE) for the C and A transitions under varying photon fluxes (Fig. S12). Notably, at the basal plane, the QE exhibited a substantial difference between the C and A transitions, depending on the presence or absence of the multilayer. Specifically, the QE for the A transition was significantly reduced in the presence of a multilayer compared to a purely monolayer region (Fig. S12a,b). In contrast, the QEs for the C transition remained largely unchanged. These findings suggest that the multilayer region impedes charge carrier extraction from bound excitons at the MoS₂-liquid interface, thereby decreasing the overall QE of the system.

Interestingly, under high photon flux, this difference became less pronounced, as spatial constraints imposed by the MoS₂ flake dimensions and the availability of charge-extracting molecules in solution remained fixed (Fig. S12c,d). The increased photon flux led to a greater number of charge generation events, which enhanced charge carrier extraction efficiency until the system reached its inherent limitations.

Fig. S12: (a,c) External and (b,d) internal quantum efficiency at different positions (basal (multi): dark red; basal (mono): grey; corner: purple) inside the ML-MoS₂ (Fig. S11) according to the excited transition at (a,b) $\sim 1.2 \times 10^{12}$ and (c,d) $\sim 14 \times 10^{12}$ photons s⁻¹ (C transition at 455 nm; A transition at 660 nm). The error bars represent the standard deviation.

To assess the role of flake size in QE measurements, we studied MoS₂ flakes of varying sizes relative to the excitation beam (approximately 80 μm²). The selected flakes ranged from 0.5 to 70 times the beam area (Fig. S13).

Fig. S13: On the left, optical images of the different flakes investigated for comparing the flake size. On the right, table containing the flake number and the corresponding area measured with ImageJ.

Measurements of EQE (Fig. S14a,c) and IQE (Fig. S14b,d) were conducted at different photon fluxes ($\sim 1.2 \times 10^{12} \text{ s}^{-1}$, Fig. S14a,b; $\sim 14 \times 10^{12} \text{ s}^{-1}$, Fig. S14c,d) at the centre of each flake, which were numbered according to their size, from the largest (Flake 1) to the smallest (Flake 8). At low photon flux, EQE and IQE remained consistent for Flakes 1 to 6 but dropped significantly for Flakes 7 and 8. This reduction was less pronounced for Flake 7 under A transition excitation. These results suggest that QE is governed by multiple factors, including flake size, excitation spot size, and the availability of charge-extracting species in solution. For flakes with an area close to the light spot size, photocarrier saturation occurred even at the lowest studied intensities. Conversely, for flakes much larger than the excitation beam (10 times or more), the maximum QE was limited by charge carrier interaction with the solution rather than by spatial confinement effects. Consequently, similar QE values were observed for all large flakes (Flakes 1 to 6).

Under higher photon flux (Fig. S14c,d), the overall trend remained unchanged, except for Flakes 5 and 6 under C transition excitation. The observed reduction in QE for these flakes suggests a transition in the limiting factor from charge carrier availability at the MoS₂-liquid interface to the MoS₂ surface area available for charge extraction.

Ultimately, our findings highlight the complexity of understanding photocatalytic behavior, even in a simple system like MoS₂. Our results underscore the importance of multiple interdependent factors in accurately assessing and comparing photocatalytic efficiency, including the availability of redox-active molecules in solution, excitation spot properties (size, wavelength, and intensity), and the MoS₂ platform characteristics (thickness, size, and defect concentration). Overall, this study presents a robust methodology for evaluating the photocatalytic properties of materials under

realistic operating conditions, allowing simultaneous assessment of both reduction and oxidation reactions without measurement-induced artifacts.

Fig. S14: (a,c) External and (b,d) internal quantum efficiency of different MoS₂ flakes (Fig. S13) according to the excited transition at (a,b) $\sim 1.2 \times 10^{12}$ and (c,d) $\sim 14 \times 10^{12}$ photons s⁻¹ (C transition at 455 nm; A transition at 660 nm). The error bars represent the standard deviation. The number of the flakes was assigned from the biggest to the smallest flakes (see Fig. S13).”

Comment 7 – Lines 258-260 “However, the EQE and IQE values decrease significantly, consistent with lateral confinement of the electrons inside the small monolayer MoS₂ flake”. It is not clear what is causing the EQE an IQE to decrease significantly? What is this statement in regards to?

Reply to Comment 7: We thank the reviewer for this question. The MoS₂ flake surface directly limits the charge carrier population. As expressed by Reviewer 3, in comment 1: “This distance is limited by the size of the triangle shown in the SI.” The size of the flake limits the charge carrier distribution. Once the MoS₂ flake is fully populated (steady state for the extraction of charge carriers at the interface), higher power density of light will only decrease the quantum efficiency. This is what we consider as lateral confinement in this sentence. We explored more in details this

aspect in the newly added Supplementary Section 3, where we discussed the effect of the MoS₂ area on the obtained QEs.

Comment 8 – The conclusions give a perspective on bilayers leading to more higher reactivity. Can the authors comment on this from their data since some of the triangles did have bilayer formation in the middle of the triangles.

Reply to Comment 8: We thank the reviewer for his/her suggestion. We modified the sentence as we did not mean bilayer of MoS₂ but the heterojunction between two different monolayer materials. We modified the text as follow:

Line 342: “This can be accomplished, for instance, by enhancing the absorption cross-sections using an underlying metasurface as a lens, and also by controlling reactive sites in 2D materials through defect engineering and the formation of **van der Waals heterostructures.**”

Reviewer #4:

The authors investigated the photochemical reactivity of 2D MoS₂ semiconductors using scanning photoelectrochemical microscopy. They proposed that photogenerated holes localize at the excitation spot, while photoelectrons exhibit high mobility and migrate away. However, similar conclusions have been validated in previous photocatalysis studies, which demonstrate that photogenerated electrons have higher mobility than holes. Additionally, while the title of the manuscript is engaging, the research content does not align with it well. Some key characterizations and discussion are missing from this article. This manuscript is recommended for rejection. Detailed comments are provided below:

We thank the reviewer for the time and thorough analysis of our manuscript. To the best of our knowledge, our study is the first to demonstrate, under working conditions, the spatially resolved mobility and reactivity of photocarriers across a single MoS₂ monolayer. We have revealed a striking contrast in the distribution of photo-reduced and photo-oxidized products, which is modulated by the position and intensity of light exposure. We hope that the revised version of our manuscript, improved based on the reviewer's valuable suggestions, will effectively convey the significance of our findings and address any concerns.

Comment 1 – The title of the work is “Spatially Resolved Photocatalytic Reactive Sites”, however, the research mainly validates that “oxidation products localize at the excitation spot, indicating stationary holes, while photoreduction occurs up to 80 microns away, showing exceptional electron mobility.” No experiments and discussion for identifying the specific oxidation sites and reduction sites are involved.

Reply to Comment 1: Our findings provide clear evidence of the substantial difference in the locations where photo-oxidation and photoreduction processes occur. Under photocatalytic conditions, charge carriers are generated at the illuminated region, in contrast to electrocatalysis, where charge carriers are supplied to the structure by the potentiostat. Our experiments demonstrated that charge carriers are extracted at specific sites: holes are extracted at the excitation spot, while electrons are extracted away from it. Furthermore, our aligned and unaligned measurements revealed that photoreduction can occur across the entire flake, including the basal plane, as shown in Fig. S9 (previously S4). These results emphasize that, during photocatalysis, the position of the light illumination also dictates the location of the photocatalytic sites, highlighting its greater significance compared to the basal plane or edge positions. This information is particularly relevant for the design of new photodevices. We reply in more details in the following comment of the Reviewer about the type of defects expected in CVD grown MoS₂.

Comment 2 – AFM measurement is suggested to perform to directly characterize the thickness of the MoS₂ nanosheets. In addition, HRTEM image should be provided to study the type of defect sites at the corner and edge of the nanosheets. Importantly, the role of these defects is suggested to be discussed.

Reply to Comment 2: We thank the reviewer for the suggestions. While we agree that AFM can provide an additional tool to characterize the thickness of the nanosheets, Raman spectroscopy has also been a well-established, reliable, and widely used optical probe to assess thickness in TMD flakes (added references 34-38). It probes it by analyzing the frequency shifts of the E_{2g} (in-plane) and A_{1g} (out-of-plane) vibrational modes, which exhibit opposing trends with increasing layer number. The E_{2g} mode redshifts due to stacking-induced modifications in intralayer bonding and long-range Coulombic interlayer interactions, while the A_{1g} mode blueshifts as additional layers enhance van der Waals restoring forces. Hence, the frequency difference between these two modes increases systematically from $\sim 19 \text{ cm}^{-1}$ (monolayer) to $\sim 25 \text{ cm}^{-1}$ (bulk), providing a non-destructive metric for determining MoS_2 thickness with atomic-layer resolution (Fig. R3). A similar trend in Raman modes can also be observed in our data in Figure 1 and Figure S2. The manuscript has been revised accordingly to include additional citations and contextualize our Raman measurements.

Line 97: “Raman spectroscopy was employed for thickness determination of the MoS_2 flakes. The E_{2g} (in-plane) and A_{1g} (out-of-plane) vibrational modes serve as reliable indicators of layer thickness, as their frequencies shift systematically with the number of layers due to changes in interlayer interactions and bonding.^{34–38} A peak difference ($\Delta_{\text{peak}} = E_{2g} - A_{1g}$) of $\sim 19 \text{ cm}^{-1}$ was observed along the flake in our sample (Fig. 1e), consistent with monolayer MoS_2 .^{35,39} Raman spectra acquired from the center of some flakes exhibited characteristic peak shifts corresponding to bilayer ($\sim 21.5 \text{ cm}^{-1}$) and multilayer ($\sim 24 \text{ cm}^{-1}$) MoS_2 (Fig. S2). These shifts, together with the contrast observed in the optical images enabled a fast identification of the presence of ML- MoS_2 , indicating only a small multilayer region to be present at the centre of the flake (Fig. 1c).”

Added references:

“34. Late, D. J., Liu, B., Matte, H. S. S. R., Rao, C. N. R. & Dravid, V. P. Rapid Characterization of Ultrathin Layers of Chalcogenides on SiO_2/Si Substrates. *Advanced Functional Materials* **22**, 1894–1905 (2012).

35. Lee, C. *et al.* Anomalous Lattice Vibrations of Single- and Few-Layer MoS_2 . *ACS Nano* **4**, 2695–2700 (2010).

36. Late, D. J. *et al.* Sensing Behavior of Atomically Thin-Layered MoS_2 Transistors. *ACS Nano* **7**, 4879–4891 (2013).

37. Parzinger, E. *et al.* Photocatalytic Stability of Single- and Few-Layer MoS_2 . *ACS Nano* **9**, 11302–11309 (2015).

38. Najmaei, S. *et al.* Vapour phase growth and grain boundary structure of molybdenum disulphide atomic layers. *Nature Mater* **12**, 754–759 (2013).”

Figure R3: Figure from ref. 35 (10.1021/nn1003937) showing dependence of layer thickness on Raman peaks' frequencies and frequency difference.

Regarding defect sites, we agree that HR-TEM and detailed defect analysis would be valuable for the structural characterization of our samples. However, such an investigation falls outside the scope of this study due to its time-intensive nature and limited relevance to our specific objectives. Extensive literature has already characterized defect sites in MoS₂ flakes (and other TMDs) using high-resolution techniques such as STM, STEM, HRTEM, and super-resolution PL microscopy. These studies consistently identify S-1 and S-2 vacancies, both of which have low formation energies, as the most prevalent defects, particularly in CVD-grown flakes like our samples, where sulfur deficiency during growth induces an n-type character. While S-vacancies are commonly observed at edges, they are equally abundant on basal planes. Additionally, edge regions often exhibit reconstructed lattice structures, including undercoordinated Mo sites due to monosulfur terminations. These defect sites have been extensively studied for their electrocatalytic activity, particularly in hydrogen evolution reaction (HER) processes.

Our study focuses on reactions occurring under light illumination and excited-state conditions, where additional factors such as charge trapping and detrapping, lattice strain, photoinduced phase transformations, and modified catalyst-adsorbate interactions can play a critical role in catalytic behavior. While MoS₂ defect sites are well understood, their behavior under photoexcitation remains largely unexplored. Capturing the dynamic structural changes induced by excited-state conditions and photocatalytic conditions would require specialized techniques such as ultrafast or in-situ TEM under illumination, which are also beyond the scope of this work. Instead, our approach directly maps activity with high spatial resolution, offering insights into reactive site distributions as a novel way for understanding photocatalytic sites under realistic conditions. Ultimately, despite potential atomic-scale defect variations among the investigated flakes, the photoreactivity consistently exhibited the same dependence on light position, wavelength, and intensity.

We added the following discussion to the manuscript reflecting the discussion above:

Line 144: “The most abundant defects in MoS₂, sulfur vacancies and edge terminations, are well-known active centres in electrocatalytic processes such as the hydrogen evolution reaction (HER).^{15,17,18,46–50} Furthermore, extensive theoretical calculations have demonstrated that the electronic structure of MoS₂ for (photo)-electrocatalytic HER is highly dependent on its phase and atomic configuration.^{51–54} However, little is known about the role of these defects in photocatalysis under excited-state conditions, where additional factors beyond static defect chemistry come into play. The obtained results deviated from the established understanding that MoS₂ exhibits catalytic activity at defect sites, specifically at the edges, for HER. Also, the measurement method itself may influence the observed activity as the proximity of the probe to the surface could introduce a hindering effect on the diffusion of the species from the MoS₂ to the solution.^{55”}

Added references:

“46. Hong, J. et al. Exploring atomic defects in molybdenum disulphide monolayers. *Nat Commun* 6, 6293 (2015).

47. Vancsó, P. et al. The intrinsic defect structure of exfoliated MoS₂ single layers revealed by Scanning Tunneling Microscopy. *Sci Rep* 6, 29726 (2016).

48. Chow, P. K. et al. Defect-Induced Photoluminescence in Monolayer Semiconducting Transition Metal Dichalcogenides. *ACS Nano* 9, 1520–1527 (2015).

49. Bretscher, H. et al. Rational Passivation of Sulfur Vacancy Defects in Two-Dimensional Transition Metal Dichalcogenides. *ACS Nano* 15, 8780–8789 (2021).

50. Li, L. et al. Role of Sulfur Vacancies and Undercoordinated Mo Regions in MoS₂ Nanosheets toward the Evolution of Hydrogen. *ACS Nano* 13, 6824–6834 (2019).

51. Kadantsev, E. S. & Hawrylak, P. Electronic structure of a single MoS₂ monolayer. *Solid State Communications* 152, 909–913 (2012).

52. Tang, Q. & Jiang, D. Mechanism of Hydrogen Evolution Reaction on 1T-MoS₂ from First Principles. *ACS Catal.* 6, 4953–4961 (2016).

53. Li, W. et al. Hydrogen evolution reaction mechanism on 2H-MoS₂ electrocatalyst. *Applied Surface Science* 498, 143869 (2019).

54. Achqraoui, M., Bekkioui, N., Jebari, H. & Ez-Zahraouy, H. Exploring layer-dependent photocatalytic and optoelectronic properties of MoS₂ for hydrogen production through DFT analysis. *Journal of Physics and Chemistry of Solids* 199, 112543 (2025).”

We also added a paragraph in the conclusion.

Line 330: “These findings reinforce that, while defects contribute to reactivity, photocatalytic efficiency is governed not only by local defect chemistry but also by optically driven excited-state properties that extend beyond the well-studied behavior of MoS₂ under electrocatalytic conditions. Crucially, by employing spatially resolved SPECM measurements, we reveal that oxidation and reduction processes are not uniformly distributed across the flake but instead correlate with distinct transport behaviors of photogenerated carriers.”

Comment 3 – In Fig. 2b, while the photogenerated holes are consumed by FcDM, what happens to the electrons? Additionally, in Fig. 2b, is hydrogen actually being produced, and if so, can it be collected?

Reply to Comment 3: We thank the reviewer for their question. We clarified the different reaction on the schemes from Fig. 2. The H₂ is not produced in that case, as no hole scavenger is available to hinder the recombination of charge carriers. Even in the case of Fig. 2c,f representing the best conditions and results, the generated H₂ is in a very low amount (ΔI of 400 fA, which is already near our limit of detection: ~100 fA, for instrumental reason).

We modified the Fig. 2 as follow:

“Fig. 6: Photocatalytic maps of oxidation and reduction products for monolayer MoS₂. **a**, Scheme depicting the aligned SPECM measurements performed on ML-MoS₂ for the photo-oxidation reaction. **(b),(c)** Illustrations describing the substrate generation – tip collection mode employed in this study for detecting the photocatalytic **(b)** oxidation and **(c)** reduction products. **d**, Optical image of ML-MoS₂ deposited on SiO₂ with the dashed red square corresponding to the investigated area in **(e)** and **(f)**. **(e),(f)** Photoactivity maps of a ML-MoS₂ (black dashed triangle)

corresponding to the highlighted flake in (d) for oxidation (e) and reduction (f) products. Conditions: $r_T = 5 \mu\text{m}$, $RG = 50$, $v_{scan} = 5 \mu\text{m s}^{-1}$, $r_{\text{Light}} = 5 \mu\text{m}$, with $E_T = -0.1 \text{ V vs Ag/AgCl}$ and $Z = 20 \mu\text{m}$ for (e), and $E_T = 0.1 \text{ V vs Ag/AgCl}$ and $Z = 5 \mu\text{m}$ for (f). The experiments were performed in aqueous solutions containing 1 mM FcDM and 0.1 M KCl (e) and 0.1 M Na_2SO_4 and 10 mM Na_2SO_3 (f).”

Comment 4 – The area of light irradiation is a crucial parameter for the photoactivity mapping test. Detailed information on this should be provided for Figures 2e–f, 3a, and 4a.

Reply to Comment 4: We thank the reviewer for this critical comment. We added the light irradiation radius on all the captions as “ $r_{\text{Light}} = 5 \mu\text{m}$ ”, when necessary.

Comment 5 – The organization and discussion of Figures 3a and 3b are difficult to follow. It is recommended to reorganize the data to enhance clarity and make it easier for readers to understand. For example, the authors discussed that “As such, a negative (positive) photoactivity should be observed when the initial concentration of redox mediator (FcDM⁺) increases (decreases), accordingly to the reaction occurring at the MoS₂-liquid interface.” Did the authors change the concentration of FcDM⁺?

Reply to Comment 5: We thank the reviewer for their suggestion. We modified the organization of the Figure 3 and the discussion to simplify the clarity of this section. Moreover, we dedicated a new supplementary section focused on the influence of the FcDM/FcDM⁺ species by reproducing experiments with fresh and 1-year-old FcDM solutions (Supplementary Section 2). We modified the manuscript as follow:

Line 167: “Taking into consideration the influence of the probe during our measurements (i.e., impeding the mass transport of the produced species),⁵⁵ we suggest that the probe itself hindered the diffusion of the generated products from the MoS₂ flake (see Supplementary Section 1). This hindering effect decreased the apparent photoactivity at the basal plane for the detection of oxidized products (Fig. 2e, Fig. S3), while it increased the concentration of reduced products during the photoreduction map at the same location (Fig. 2f, Fig. S4).

To verify the distribution of the reactive sites along the MoS₂ flake, we performed spatially-resolved investigations with aligned and unaligned excitation/detection positions to determine the underlying mechanism behind our observations (Fig. 2e,f). The measurements were performed with a nanoprobe at a distance ($Z = 5 \mu\text{m}$) larger than two times the radius of the probe insulating part (r_{ins}) avoiding the hindering effect of the probe on the diffusion of species. The FcDM/FcDM⁺ redox couple was used to decipher the redox reaction occurring at the probe position. Thus, the probe was positioned away from the excitation spot (Fig. 3a) or, alternatively, aligned with it (Fig. 3b). The oxidation (reduction) process occurs when the photogenerated hole (electron) is extracted from the MoS₂ by the surrounding molecules at the MoS₂-liquid interface. As such, a negative (positive) photoactivity should be observed when the initial concentration (C_{Dark}) of redox mediator (FcDM⁺) increases (decreases), accordingly to the reaction occurring at the MoS₂-liquid interface under illumination (C_{Light}).

Interestingly, the measured photoactivity was positive (reduction) when the probe was away from the excitation spot (Fig. 3c), but always negative (oxidation) when the probe was aligned with the light spot (Fig. 3d). In both cases, a rapid steady-state was observed revealing no influence from diffusion of the species.⁵⁶ This shows that the photogenerated charge carriers were extracted at the reactive sites in accordance with the observed redox process: holes at the light excitation spot, and electrons away from the light excitation spot (Fig. 3e). Since the sign of ΔI remained consistent across all measurements at a single probe position, this suggests that, regardless of the excited optical transitions, the holes are relatively static, while the electrons exhibit significantly higher mobility, traveling through the ML-MoS₂. These results provide insight into our previous observations for the aligned measurements, which only detected the products from photo-oxidation (Fig. 2e) or photoreduction (Fig. 2f) diffusing to the probe. This confirmed a significant charge carrier separation, where hole extraction is relatively localized at the excitation area, and electrons exhibit high mobility throughout the ML-MoS₂ (Fig S4b). Furthermore, this corroborates our discussions on the hindrance of the probe on the diffusion of photoproducts from the MoS₂ to the probe (Supplementary Section 1), reinforcing the spatially selective nature of the observed photoprocesses (Fig. 3e).”

We modified the Fig. 2 as follow:

“Fig. 7: Photocatalytic maps of oxidation and reduction products for monolayer MoS₂. a, Scheme depicting the aligned SPECM measurements performed on ML-MoS₂ for the photo-oxidation reaction. **(b),(c)** Illustrations describing the substrate generation – tip collection mode employed in this study for detecting the photocatalytic **(b)** oxidation and **(c)** reduction products. **d,** Optical image of ML-MoS₂ deposited on SiO₂ with the dashed red square corresponding to the investigated area in **(e)** and **(f)**. **(e),(f)** Photoactivity maps of a ML-MoS₂ (black dashed triangle)

corresponding to the highlighted flake in (d) for oxidation (e) and reduction (f) products. Conditions: $r_T = 5 \mu\text{m}$, $RG = 50$, $v_{scan} = 5 \mu\text{m s}^{-1}$, $r_{\text{Light}} = 5 \mu\text{m}$, with $E_T = -0.1 \text{ V vs Ag/AgCl}$ and $Z = 20 \mu\text{m}$ for (e), and $E_T = 0.1 \text{ V vs Ag/AgCl}$ and $Z = 5 \mu\text{m}$ for (f). The experiments were performed in aqueous solutions containing 1 mM FcDM and 0.1 M KCl (e) and 0.1 M Na_2SO_4 and 10 mM Na_2SO_3 (f).”

We revised Fig. 3 with additional schemes for clarity:

“**Fig. 8: Aligned-unaligned excitation-detection experiments in monolayer MoS₂.** (a),(b) Illustrations depicting the measurements performed with the nanoprobe when the excitation – detection was unaligned (a), or aligned (b). (c),(d) Photoactivity (ΔI) measured far from the light excitation spot (c), and aligned with the light excitation spot (d), while the ML-MoS₂ was excited under chopped light (5 s; increased light power at every pulse; transition C, 455 nm, purple; exciton B, 595 nm, yellow; exciton A, 660 nm, dark red). The probe positions correspond to the respective inset image, showing the monolayer (ML-) MoS₂ flake (white dashed triangle), the position of the light excitation spot (black dashed circle), and the ultramicroelectrode (UME) position (red dashed circle). e, Scheme representing the spatial distribution of the photo-redox processes detected at the UME in ML-MoS₂ when excited at a fixed position, with the corresponding photogenerated charge carriers (oxidation: holes, blue region; reduction: electrons, red region). Conditions: $r_T = 0.1 \mu\text{m}$, $RG = 25$, $r_{\text{Light}} = 5 \mu\text{m}$ and $Z = 5 \mu\text{m}$ for (c),(d). The experiments were performed in aqueous solutions containing 1 mM FcDM and 0.1 M KCl. FcDM⁺ is present in low concentration due to the redox couple equilibrium in water (Fig. S5).”

We added the Supplementary Section 2 as follow:

Line 105 in SI:

“Supplementary Section 2: Influence of the FcDM⁺ species on the observed photoactivity.

To better illustrate the effect of the FcDM⁺ species on system limitations, we performed aligned-unaligned excitation-detection measurements on freshly prepared and one-year-old 1 mM FcDM solutions. Due to the equilibrium between FcDM and FcDM⁺ in water, the aging process resulted in a shift toward a lower FcDM concentration (and a correspondingly higher FcDM⁺ concentration) in the one-year-old solution (Fig. S6).

Fig. S6: Cyclic voltammetry of 1 mM FcDM solution freshly prepared (dark) and 1-year-old (purple) representing the normalized probe current according to the probe current obtained at the oxidation peak for the 1 mM FcDM fresh solution.

This decrease in the reduced species (FcDM) and corresponding increase in the oxidized species (FcDM⁺) influence the extraction efficiency of photogenerated charge carriers during the measurements. The relative signal intensity provides direct information on species concentration, as only the species concentration varies (Eq. S7). Considering a normalized probe current of 1 for the fresh 1 mM FcDM solution, the FcDM⁺ species exhibited a normalized intensity of 0.06. In contrast, the one-year-old solution yielded normalized intensities of 0.78 and 0.15 for FcDM and FcDM⁺, respectively.

As in Fig. 3, we conducted aligned-unaligned excitation-detection measurements for both solutions and compared the results (Fig. S7). Interestingly, the aligned measurements exhibited similar photoactivities. However, the one-year-old solution displayed a slightly more negative photoactivity for both C and A transitions. The most significant difference emerged in measurements taken away from the illumination position (as indicated in Fig. S7a). While the fresh FcDM solution exhibited a signal comparable to that shown in Fig. 3, no detectable signal was observed for the one-year-old FcDM solution (Fig. S7b,c). This can be attributed to the changes in the availability of FcDM/FcDM⁺ species: the concentration of FcDM molecules available for hole extraction decreased by approximately 20%, whereas the availability of FcDM⁺ for electron extraction increased by nearly 200%. Consequently, the local concentration of FcDM⁺ became less limiting compared to the fresh FcDM solution.

Fig. S7: a, Optical image showing the monolayer (ML-) MoS₂ flake, the position of the light excitation spot, and the ultramicroelectrode (UME) position corresponding to the unaligned excitation-detection measurements. For the aligned excitation-detection measurements, the UME was positioned at the light excitation spot. **(b,c)** Photoactivity measurements under chopped light excitation (5 s; increased light power at every pulse) for the aligned (negative ΔI) and unaligned (positive ΔI) measurements for the C transition **(b)** and the A transition **(c)** performed in presence of fresh 1 mM FcDM (black curves) and one-year-old 1 mM FcDM (colored curves) solutions. Conditions: $r_T = 0.1 \mu\text{m}$, $R_G = 25$, and $Z = 5 \mu\text{m}$ for **(b,c)**. The initial solutions were prepared with 1 mM FcDM and 0.1 M KCl.

To further investigate the regions where reduction products formed on the MoS₂ flake immersed in the one-year-old FcDM solution, we performed unaligned excitation-detection measurements at different locations along the flake (Fig. S8). We selected four positions for probing photoproducts: (1) far from the illumination site ($\sim 75 \mu\text{m}$), (2) mid-distance ($\sim 30 \mu\text{m}$), (3) near the illumination site ($\sim 10 \mu\text{m}$), and (4) directly adjacent to the illuminated region ($\sim 1\text{--}2 \mu\text{m}$) (Fig. S8a). Notably, photoactivity associated with reduced products peaked at position 3 for the C transition (Fig. S8b) and at position 4 for the A transition (Fig. S8c). A sharp increase in signal intensity was observed immediately upon illumination, followed by a decrease, suggesting that reduction occurs instantaneously while oxidized products diffuse from the excitation site toward the probe (Fig. S8b). This effect was particularly pronounced for the C transition at position 4, where the high diffusion of FcDM⁺ resulted in a decrease in photoactivity, ultimately yielding negative ΔI values. In contrast, the species generated under excitation at the A transition exhibited little to no diffusion. This behavior may be attributed to differences in carrier dynamics, as A-excitons correspond to more localized and tightly bound carriers compared to those associated with the C transition. Furthermore, the absence of photoactivity at position 1 indicates that electrons no longer reached this region. We hypothesize that the combination of the excitation spot size and the decrease in FcDM concentration acted as limiting factors in the one-year-old FcDM solution. This highlights the critical influence of electrolyte composition (availability of species to be reduced and oxidized) and light excitation position on the photoreaction dynamics occurring at the MoS₂ platform.

Fig. S8: **a**, Optical image showing the monolayer (ML-) MoS₂ flake, the position of the light excitation spot, and the ultramicroelectrode (UME) positions corresponding to the unaligned excitation-detection measurements performed at the different positions indicated on the image: (1) far from the illumination site (~75 μm), (2) mid-distance (~30 μm), (3) near the illumination site (~10 μm), and (4) directly adjacent to the illuminated region (~1–2 μm). **(b,c)** Photoactivity measurements under chopped light excitation (5 s; increased light power at every pulse) at the different positions indicated in **(a)** for the C transition **(b)** and the A transition **(c)** performed in presence of one-year-old 1 mM FcDM solutions. Conditions: $r_T = 0.1 \mu\text{m}$, $RG = 25$, and $Z = 5 \mu\text{m}$ for **(b,c)**. The initial solution was prepared with 1 mM FcDM and 0.1 M KCl.”

Comment 6 – According to Figure 2b and c, “the probe electrochemically detects the oxidized (negative ΔI) or reduced (positive ΔI) products generated from the photocatalytic reactions occurring at the ML-MoS₂-liquid interface using different electrolytes”. However, both negative and positive current are detected in Figure 3b using FcDM electrolyte. More discussion for the reaction process or chemical equation is suggested to provide.

Reply to Comment 6: We thank the reviewer for highlighting this point. We added the different reactions in a new Supplementary Section 1, which focuses on the explanation of the SPECM measurements presented in figure 2. We also added new schemes in Fig. 3 to increase the clarity of our manuscript. Moreover, we added an example of CV for fresh and old FcDM solutions, showing the difference in ratio of FcDM/FcDM⁺ species due to the equilibrium shifting during the aging process (Fig. S6).

We added the corresponding equations for the reactions occurring during the experiments for both scenarios in the new Supplementary Section 1:

“The following reactions occurred during the photo-oxidation map:

Electrochemical reduction at the probe: $FcDM^+ + e^- \rightarrow FcDM$ Eq. S3”

“The following reactions occurred during photoreduction experiments:³

Photo-oxidation reaction at the MoS₂ surface: $SO_3^{2-} + 2h^+ + H_2O \rightarrow SO_4^{2-} + 2H^+$ Eq. S4

Photo-reduction reaction at the MoS₂ surface: $2H^+ + 2e^- \rightarrow H_2$ Eq. S5

Electrochemical oxidation at the probe: $H_2 \rightarrow 2H^+ + 2e^-$ Eq. S6”

Revised Fig. 3 with the additional schemes for clarity:

“**Fig. 9: Aligned-unaligned excitation-detection experiments in monolayer MoS₂.** (a),(b) Illustrations depicting the measurements performed with the nanoprobe when the excitation – detection was unaligned (a), or aligned (b). (c),(d) Photoactivity (ΔI) measured far from the light excitation spot (c), and aligned with the light excitation spot (d), while the ML-MoS₂ was excited under chopped light (5 s; increased light power at every pulse; transition C, 455 nm, purple; exciton B, 595 nm, yellow; exciton A, 660 nm, dark red). The probe positions correspond to the respective inset image, showing the monolayer (ML-) MoS₂ flake (white dashed triangle), the position of the light excitation spot (black dashed circle), and the ultramicroelectrode (UME) position (red dashed circle). e, Scheme representing the spatial distribution of the photo-redox processes detected at the UME in ML-MoS₂ when excited at a fixed position, with the corresponding photogenerated charge carriers (oxidation: holes, blue region; reduction: electrons, red region). Conditions: $r_T = 0.1 \mu\text{m}$, $RG = 25$, $r_{\text{Light}} = 5 \mu\text{m}$ and $Z = 5 \mu\text{m}$ for (d),(e). The experiments were performed in aqueous solutions containing 1 mM FcDM and 0.1 M KCl. FcDM⁺ is present in low concentration due to the redox couple equilibrium in water (Fig. S5).”

Manuscript: Spatially Resolved Photocatalytic Reactive Sites and Quantum Efficiency in a 2D Semiconductor - NCOMMS-24-67854A

REPLY TO THE REVIEWERS' COMMENTS

We thank the editor and the reviewers for the time spent in the revision of our manuscript. We address below point-by-point the Reviewers' comments. Please find our responses in red and highlighted in yellow the changes introduced to the manuscript and supplementary information.

Reviewer #1:

The authors' response letter provides adequate experimental supplementation and literature support. However, further clarifications are required regarding: Comparative data for innovation validation, Dynamic tracking of reaction sites, Quantitative mechanisms of multilayer/defect effects, Independent verification of H₂ production. Addressing these issues will significantly enhance the paper's rigor and academic impact. Please reference the following comments: Questions Requiring Further Clarification and Improvement Suggestions

Reply to Reviewer: We thank the reviewer for their time and continued engagement with our manuscript. While we appreciate the suggestions provided, we note that the points raised in this round were already addressed in detail during the previous revision. Our responses were acknowledged by Reviewers 2, 3, and 4 as satisfactory (comments 2, 3, 4, 5, and 7), and no further concerns on these aspects were raised at that stage. We trust that the updated manuscript, when considered alongside the full supplementary materials, clearly conveys our methodology, results, and interpretations. Nonetheless, we have revisited each of the comments again here to ensure that our rationale remains transparent and well supported.

Comment 1 – Insufficient Innovation (Reviewer 1, Comment 3)
Issue: The authors emphasize methodological innovation (e.g., regulating reaction sites via photoexcitation positioning) but lack direct comparisons with existing technologies.

Recommendations:

Add comparative experiments with electrocatalytic or traditional photocatalytic systems to demonstrate the unique advantages of photoexcitation positioning. Include comparisons with literature-reported carrier mobility data (e.g., PL lifetime or transient absorption spectroscopy) to validate the uniqueness of long-range electron migration.

Reply to Comment 1: We respectfully note that our findings do not lie in benchmarking but in the spatial decoupling of photoinduced oxidation and reduction reactions via excitation positioning, a phenomenon we observed in operando via SPECM. As previously discussed, we compare aligned-unaligned excitation-detection measurements on identical flakes. Additional benchmarking against electrocatalytic systems or PL-based mobility values is beyond the scope of our study. Furthermore, Reviewers 2, 3 and 4 have already recognized the novelty of our approach and the significance of our results in elucidating the spatial distribution of photogenerated charge carriers and their reactivity on a monolayer MoS₂ flake.

Comment 2 – Dynamic Tracking of Reaction Sites (Reviewer 2, Comment 1)
Issue: The response lacks direct dynamic evidence for how photoexcitation positioning governs reaction sites.

Recommendations: Incorporate time-resolved SPECIM imaging to visualize the diffusion pathways of oxidation/reduction products after photoexcitation. Use ultrafast spectroscopy (e.g., transient fluorescence) to study carrier migration kinetics and clarify the regulatory mechanism of photoexcitation positioning on reaction sites.

Reply to Comment 2: As presented in the manuscript and Supplementary Information, our chopped-light SPECIM experiments provide time-resolved insight into photoactivity and steady-state response under repeated illumination cycles (Fig. 3c–d, S9, and S10). These measurements directly assess the spatial and temporal evolution of redox-active species, allowing us to identify and quantify instances where diffusion occurs, which we explicitly discuss. Therefore, our measurements unambiguously reveal distinct spatial distributions of photocarrier populations at and away from the illumination spot.

Line 151 in Supplementary Information – Section 2: “A sharp increase in signal intensity was observed immediately upon illumination, followed by a decrease, suggesting that reduction occurs instantaneously while oxidized products diffuse from the excitation site toward the probe (Fig. S10b). This effect was particularly pronounced for the C transition at position 4, where the high diffusion of FcDM^+ resulted in a decrease in photoactivity, ultimately yielding negative ΔI values.”

Comment 3 – Quantitative Impact of Multilayer Structures (Reviewer 3, Comment 6)
Issue: Supplementary Section 3 only qualitatively discusses the effect of multilayer regions on quantum efficiency, lacking quantitative mechanistic analysis.
Recommendations:

Conduct comparative experiments on MoS_2 with varying layer numbers (monolayer, bilayer, multilayer) to establish a quantitative thickness-efficiency relationship. Utilize scanning Kelvin probe microscopy (SKPM) or surface photovoltage (SPV) techniques to investigate the influence of multilayer regions on charge separation efficiency.

Reply to Comment 3: We thank the reviewer for their suggestion. However, we would like to clarify that our study does not aim to establish a thickness-efficiency relationship. Our flake selection was based on optically verified CVD-grown monolayers, some of which contained small, unintentional multilayer regions. As detailed in Supplementary Section 3, our goal was to rule out multilayer-induced effects as the origin of the observed spatial variations in quantum efficiency. By comparing EQE and IQE across regions with and without multilayer domains (Fig. 4b,c and Fig. S14), we show that the observed changes in QE cannot be attributed to thickness variation.

Comment 4 – Direct Characterization of Defect Sites (Reviewer 4, Comment 2)
Issue: Defect types are indirectly inferred from literature without direct experimental characterization of the sample.

Recommendations:

Perform atomic-level characterization (e.g., STM or HAADF-STEM) to directly observe S-vacancy or edge defect distributions.

If experimental constraints exist, at least employ PL spectroscopy or Raman mapping (e.g., defect-related peak intensity) to indirectly assess defect density.

Reply to Comment 4: We thank the reviewer for their comment. We note that the evolution of photoactivity observed in our SPECM measurements follows the *excitation spot*, rather than the edges or other static defect-related regions. This shows that spatial reactivity is governed by photoinduced carrier dynamics rather than static defect distributions.

While we acknowledge that atomic-scale techniques such as STM or HRTEM can directly visualize defects, they are not compatible with the aqueous, in-operando conditions of our study and would not reflect the material's functional state during photocatalysis. As such, they are not appropriate for the scope of this work.

Nonetheless, to address the reviewer's concern, we have now included photoluminescence (PL) peak intensity mapping of the investigated MoS₂ flakes (Fig. S12 and S13). This provides spatially resolved information on defect-related emission and are widely used and accepted for characterizing 2D materials in conditions relevant to our study.

Revised Fig. S12 and S13:

Fig. S12: **a**, Optical image of Flake 1 and Flake 2 from Fig. 4, highlighting the presence of a multilayer at the centre of Flake 1. **b**, Normalized PL peak intensity map of the MoS₂ flakes in (a).

Fig. S13: **a**, Optical image corresponding to the investigated monolayer MoS₂ flake with the different positions where measurements were performed: purple for corner, dark red for basal at the centre with the presence of multilayer (namely, basal (multi)), and grey for basal away from the centre to avoid any multilayer (namely, basal (mono)). **b**, Normalized PL peak intensity map of the MoS₂ flake in (a). The inset shows the high-resolution PL peak intensity map of the region containing multilayers.

Comment 5 – Independent Validation of H₂ Detection (Reviewer 2, Comment 2)
Issue: H₂ detection relies solely on electrochemical probes, risking interference from other reduction products.

Recommendations:

Supplement with online mass spectrometry (DEMS) or gas chromatography (GC) to independently verify H₂ generation. Compare results under sodium thiosulfate-free conditions to exclude interference from competing reduction reactions (e.g., O₂ reduction).

Reply to Comment 5: We thank the reviewer for raising this point. The H₂ signal is detected using a platinum ultramicroelectrode biased at 0.1 V vs. Ag/AgCl, a potential where H₂ oxidation is well established and selective (DOI: 10.1021/jacs.3c00663; 10.1021/acs.jpcc.5b03511).

(i) O₂ reduction does not occur at this potential (if it did, we would have a negative I_{Tip} as a dark current, Fig. R1).

(ii) H₂O₂, if present, would be reduced and result in a negative ΔI, which was not observed (DOI: 10.1016/j.elecom.2013.04.014).

(iii) The 10 mM Na₂SO₃ used as a hole scavenger is also a known O₂ scavenger, present at concentrations far exceeding O₂ solubility, further eliminating oxygen-related interference.

(iv) If any residual Na₂SO₃ oxidation were occurring at the probe, it would contribute to the dark current and manifest as a decrease in signal under illumination.

Considering these factors, the measured current arises specifically from H₂ oxidation.

Figure R1. Probe current maps corresponding to the activity map without light illumination and corresponding to the map presented in Fig. 2f in the main text. Conditions: $r_T = 5 \mu\text{m}$, $RG = 50$, $v_{scan} = 5 \mu\text{m s}^{-1}$, $r_{\text{Light}} = 5 \mu\text{m}$, with $E_T = 0.1 \text{ V}$ vs Ag/AgCl and $Z = 5 \mu\text{m}$.

Comment 6 – Originality of Theoretical Calculations (Reviewer 1, Comment 2)
Issue: Theoretical analysis cites existing frameworks without customized calculations aligned with experimental findings.

Recommendations:

Perform DFT-based simulations of carrier migration pathways to explain long-range electron transfer mechanisms.

Calculate charge separation efficiency differences under varying excitation wavelengths (A/C excitons) to correlate with experimental results.

Reply to Comment 6: We thank the reviewer for the suggestion. As previously addressed, our focus is experimental, with mechanistic insights derived from spatial photoactivity mapping.

We explicitly referenced multiple relevant DFT studies (Refs. 51–54) to contextualize the role of phase, edge vs basal sites, and exciton localization. To our knowledge, DFT cannot capture electron migration over tens of micrometer scales and under non-equilibrium photocatalytic conditions.

Comment 7 – Title-Content Consistency (Reviewer 4, Comment 1)
Issue: The title emphasizes "atomically resolved photocatalytic active sites," yet the work fails to explicitly reveal atomic-level site information.

Recommendations:

Reply to Comment 7: We thank the reviewer for their comment. However, we would like to clarify that our title does not include the word “atomically”, and we do not claim atomic-scale identification of reactive sites.

Reviewer #2:

In the present manuscript, Henrotte et al. employed a combined photoelectrochemical method to study the spatial distribution of photogenerated charge carriers and their reactivities on a prototypical 2D semiconductor monolayer (ML-MoS₂). Following a thorough review of the revised manuscript, it is evident that the authors have meticulously addressed all the concerns raised, particularly the experimental details pertaining to the SPECM measurements. In light of these revisions, I believe that this version has met the requisite standards of Nature Communications and can be published without a need of further revision.

Reply to Reviewer: We thank the reviewer for their thoughtful evaluation and for recognizing the efforts we made to address all prior concerns. We greatly appreciate the positive feedback on the quality of our revisions and the clarity of the experimental framework. We are pleased that the current version of the manuscript meets the reviewer's expectations, and we thank them for recommending it for publication.

Reviewer #3:

The authors have added a lot of detail to their manuscript and SI which is helpful. However, I still think that more is needed to understand the difference in the photooxidation/reduction mapping to differentiate what is real and what is due to probe hinderance.

Reply to Reviewer: We thank the reviewer for acknowledging the improvements made to the manuscript and Supplementary Information. We appreciate the question on the probe hindrance and understand the need for further clarifications. We hope that the revised version resolves the remaining concerns.

Comment 1 – However, I am still confused by their response to the probe hinderance and how that is impacting their results. To help with this, I would suggest that the authors perform the same measurements looking at the oxidation and reduction of FcDM to convince the reader of their conclusions. By switching chemical reactions to probe reduction and oxidation, they authors add confusion about their results and the differences in photomapping.

Furthermore, I do not find the Figure S3 and S4 schematics helpful in describing the challenges with the probe hinderance and suggest they prepare different schematics to illustrate their point.

Reply to Comment 1: We thank the reviewer for their comment. We agree that the switching of redox reactions between Fig. 2e and f may initially introduce some confusion. However, we needed this configuration, as this contrast was key to uncovering a central insight of the study (the influence of probe hindrance in revealing spatially separated charge carrier dynamics).

Aligned SPECM measurements using FcDM for both oxidation and reduction would yield similar ΔI profiles, since the underlying redox processes at the MoS₂ surface are the same. This has been demonstrated in prior SPECM studies using redox mediators (e.g., 10.1039/C8FD00057C; 10.1021/acsnano.9b00219; 10.1021/acsnano.3c01009; 10.1021/acs.nanolett.4c01386). Instead, the use of water reduction in the presence of Na₂SO₃ allows for selective removal of photoinduced holes and enables specific observation of electron-related activity. This distinction would not be accessible by probing FcDM alone for both aligned photo-oxidation and photoreduction.

In the case of H₂ detection, the observed signal appears strongest at the basal plane. At first glance, this contradicts the expectation that hydrogen evolution should be localized at known reactive sites (e.g., edges or defect-rich regions). However, this apparent contradiction arises precisely because of the probe's geometry: the large insulating sheath regulates mass transport from the flake to the probe (e.g., probe area ~2000 μm²; flake area ~700 μm²). Under these conditions, higher signal from H₂ is detected when more of the flake is covered by the probe.

Therefore, the enhanced H₂ signal at the basal plane is not indicative of local hydrogen production, but rather of electron migration away from the illumination site, followed by reduction at spatially distant regions, which is made visible due to the diffusion hindrance imposed by the probe. If H₂ were generated solely and locally at the excitation spot, the probe would not affect the measured

current and no significant differences would be detected along the flake. This indirect evidence strongly suggests our interpretation of long-range electron migration, that we later demonstrate with the aligned-unaligned excitation-detection measurements.

Regarding Fig. S3 and S4, our intent was to schematically highlight the probe hindrance mechanism in a simplified form, we agree that the current version could be improved for clarity. In the revised version, we have included new schematic illustrations based on simplified COMSOL simulations that explicitly show species generation and diffusion under different probe and light configurations. We revised the section accordingly including these additional figures. We hope that the revised version better reflect the hindering effect of the probe on the mass transport.

Line 48 in the Supplementary Information: “Due to the counter reaction occurring at the MoS₂ flake and the hindrance caused by the probe, the apparent activity of the basal plane (Fig. S3a) and the corner (Fig. S3b) are influenced by the surrounding species produced away from the excitation spot, since in every position in the solution the sum of FcDM and FcDM⁺ molecules remain constant.”

Revised Figure S3:

Fig. S3: Schemes describing the situation where the probe is detecting the species at the centre (a) or at the edge (b) of the MoS₂ flake during the photo-oxidation map. The probe size, the light illumination spot, and the distance between the probe and the investigated material correspond to the conditions of the photo-oxidation map presented in Fig. 2e, according to the scalebar of 5 μm.

Line 64 in the Supplementary Information: “We performed the measurements in the presence of a hole scavenger to enhance the photogenerated electron lifetime, and remove the contribution of the oxidized species on the measured current at the probe.”

Revised Figure S4:

Fig. S4: Schemes describing the proposed scenarios, where the probe detects the species generated directly at the basal plane due to the photoreduction process occurring locally at the light excitation (a), or the probe detects the species generated at the edge of the MoS₂ flake due to the electron migration to the reactive sites and the subsequent hindered diffusion of the species related to the probe size (b). The probe size, the light illumination spot, and the distance between the probe and the investigated material represent the conditions during to the photoreduction map presented in Fig. 2f, according to the scalebar of 5 μm.

Line 97 in the Supplementary Information: “To understand the effect of the probe hindrance on the generation of molecules according to the probe’s position, we simulated the diffusion of molecules from the surface considering their production at the light position (Fig. S5), as in Fig. S4a. We observe similar diffusion profile independently of the position in the flake as the irradiated area remain the same, which does not correspond to the experimental results obtained in Fig. 2f.

Fig. S5: Simulation model results obtained for the production of H₂ at the light excitation spot in presence of the probe aligned with the light positioned at the corner (a) or at the centre (b) of the MoS₂ flake. The probe size, the light illumination spot, and the distance between the probe and the investigated material represent the conditions during to the photoreduction map presented in Fig. 2f, according to the scalebar of 10 μm.

To understand the hindering effect from the probe on the measurements, a dimensionless parameter has been previously introduced as L ,⁵ corresponding to the ratio between the probe-

substrate distance (Z) and the active part radius (r_T). For $L \geq 10$, no hindering effect is expected. For the measurements in Fig. 2f, $L = 1$, which suggests a significant hindrance from the system on the diffusion, forcing the molecules to diffuse from the MoS₂ to the probe. To highlight the hindrance of the probe on the generated species, we simulated with the same model the scenario represented in Fig. S4b in absence (Fig. S6a,c) and presence of the electrochemical probe (Fig. S6b,d) at the centre and the corner of the MoS₂ flake. This reveals i) the significant hindrance of the probe on the diffusion of H₂ in solution, and ii) a similar trend than the one observed in Fig 2f with H₂ detected at the probe in higher quantity at the centre than the corner of the MoS₂ flake (Fig. S6b,d). Furthermore, we believe that the detection of H₂ in this system was possible due to the probe hindrance, as bare MoS₂ exhibits significantly low photogeneration of H₂.

Fig. S6: Simulation model results obtained for the production of H₂ away from the light excitation spot without the probe (a,c) or with the probe aligned with the light (b,d) positioned at the centre (a,b) or at the corner (c,d) of the MoS₂ flake. The probe size, the light illumination spot, and the distance between the probe and the investigated material represent the conditions during the photoreduction map presented in Fig. 2f, according to the scalebar of 10 μm .”

In the method section: “*Simulation model:* A two-dimensional diffusion model was implemented in COMSOL Multiphysics (v6.1) using the transport of diluted species physics interface to simulate the steady-state spatial distribution of molecular hydrogen in aqueous solution, generated heterogeneously from a reactive surface. The objective was to evaluate local concentration gradients and the impact of mass transport limitations in the presence of a non-reactive structure mimicking our electrochemical probe.

The simulation domain measured 60 μm in width and 20 μm in height. A 40 μm -wide reactive surface was positioned along the bottom edge of the domain, centered horizontally. To represent the physical presence of the probe, a 50 μm -wide passive object was placed 5 μm above the surface, aligned with the center of the illuminated region. Hydrogen generation was considered under two distinct configurations: (i) localized generation within a 10 μm -wide region centered beneath the probe, corresponding to direct photoexcitation at the light spot, and (ii) distributed

generation over the remaining 30 μm of the reactive surface, simulating activity occurring away from the illuminated region.

In each case, an arbitrary total flux was imposed and normalized over the respective generation zone to maintain consistent overall H_2 production. The diffusion coefficient of hydrogen was set to $4.8 \times 10^{-9} \text{ m}^2 \cdot \text{s}^{-1}$, and the initial concentration of H_2 was assumed to be zero throughout the domain.”

Comment 2 – A new paper: DOI: 10.1021/acsnano.4c13276 has been published and could be helpful in the introduction to motivate this project.

Reply to Comment 2: We thank the reviewer for pointing out the newly published work (DOI: 10.1021/acsnano.4c13276). We have reviewed this article and find it a valuable contribution. We have now included a citation to this work in the revised Introduction and used it to help frame the motivation for spatially resolved photocatalytic investigations in 2D materials.

Line 66: “Therefore, it is essential to identify active sites and monitor their reactivity under appropriate *operando* conditions, using advanced approaches that can probe these aspects microscopically. A recent study highlighted the power of such *operando* approaches by revealing the local MoS_2 photoreactivity according to the layer number.²² These insights are critical for providing a deeper understanding that will lead to novel photocatalyst design, fully leveraging the tunability of 2D TMDs to develop next-generation materials and set new benchmarks in efficiency and sustainability.⁸”

Added reference: “22. Bo, T., Ghoshal, D., Wilder, L. M., Miller, E. M. & Mirkin, M. V. High-Resolution Mapping of Photocatalytic Activity by Diffusion-Based and Tunneling Modes of Photo-Scanning Electrochemical Microscopy. *ACS Nano* 19, 3490–3499 (2025).”

Reviewer #4:

The revised manuscript is suggested to be published.

Reply to Reviewer: We sincerely thank the reviewer for supporting the publication of our revised manuscript.

Manuscript: Spatially Resolved Photocatalytic Reactive Sites and Quantum Efficiency in a 2D Semiconductor - NCOMMS-24-67854B

REPLY TO THE REVIEWERS' COMMENTS

We thank the editor and the reviewers for the time spent in the revision of our manuscript. We address below the Reviewers' comments. Please find our responses in red.

Reviewer #1:

It seems OK and to be accepted. Anyways, the date is indeed insufficient. If the authors add more date, it can be awesome.

Reply to Reviewer: We are pleased that the current version of the manuscript meets the reviewer's requirements, and we thank them for recommending it for publication.

Reviewer #3:

The reviewers have addressed my comments and I support publishing.

Reply to Reviewer: We thank the reviewer for supporting the publication of our revised manuscript.